# MASK AND RESTORE: BLIND BACKDOOR DEFENSE AT TEST TIME WITH MASKED AUTOENCODER

## ABSTRACT

Deep neural networks are vulnerable to backdoor attacks, where an adversary maliciously manipulates the model behavior through overlaying images with special triggers. Existing backdoor defense methods often require accessing a few validation data and model parameters, which are impractical in many real-world applications, *e.g.*, when the model is provided as a cloud service. In this paper, we address the practical task of blind backdoor defense at test time, in particular for black-box models. The true label of every test image needs to be recovered on the fly from a suspicious model regardless of image benignity. We focus on test-time image purification methods that incapacitate possible triggers while keeping semantic contents intact. Due to diverse trigger patterns and sizes, the heuristic trigger search in image space can be unscalable. We circumvent such barrier by leveraging the strong reconstruction power of generative models, and propose a framework of *Blind Defense with Masked AutoEncoder* (BDMAE). It detects possible triggers in the token space using image structural similarity and label consistency between the test image and MAE restorations. The detection results are then refined by considering trigger topology. Finally, we fuse MAE restorations adaptively into a purified image for making prediction. Our approach is blind to the model architectures, trigger patterns and image benignity. Extensive experiments under different backdoor settings validate its effectiveness and generalizability.

## 1 INTRODUCTION

Deep neural networks have been widely used in various computer vision tasks, like image classification (Krizhevsky et al., 2012), object detection (Girshick et al., 2014) and image segmentation (Long et al., 2015), *etc*. Despite the superior performances, their vulnerability to backdoor attacks has raised increasing concerns (Gu et al., 2019; Nguyen & Tran, 2020; Turner et al., 2019). During training, an adversary can maliciously inject a small portion of poisoned data. These images contain special triggers that are associated with specific target labels. At inference, the backdoored model behaves normally on clean images but makes incorrect predictions on images with triggers.

To defend against backdoor behaviors, existing methods often require accessing a few validation data and model parameters. Some works reverse-engineer triggers (Wang et al., 2019; Guan et al., 2022), and mitigate backdoor by pruning bad neurons or retraining models (Liu et al., 2018; Wang et al., 2019; Zeng et al., 2021). The clean labeled data they require, however, are often unavailable. A recent work shows that the backdoor behaviors could be cleansed with unlabeled or even out-of-distribution data (Pang et al., 2023). Instead of modifying the model, Februus (Doan et al., 2020) detects triggers with GradCAM (Selvaraju et al., 2017), and feeds purified images to the model.

All these defending methods, although effective, assume the model is known. Such white-box assumption, however, may be violated in many real-world scenarios. Due to increasing concerns on data privacy and intellectual property, many models are provided as black-boxes where detailed parameters are concealed (Dong et al., 2021; Guo et al., 2022; Chen et al., 2019), *e.g.*, a cloud service API. It is thus crucial to address the problem for black-box models.

In this paper, we tackle the extreme setting and address the task of *Blind Backdoor Defense at Test Time*, in particular for black-box models. *Blind* means that there is no information on whether the model and test images are backdoored or not. Shown in Fig. 1, the prediction model is black-box and may have been injected a backdoor. Test images come in a data stream. The true label of each

Figure 1: Test time blind backdoor defense with a black-box prediction model (may be backdoored). Test images come in a stream. The defender purifies them to predict the correct labels on-the-fly.

test image is unknown; it needs to be recovered on the fly only from the hard label predictions of the suspicious model, without accessing additional data. This is a very challenging task that cannot be solved by existing test-time defense methods. Simply applying test-time image transformations (Gao et al., 2019; Sarkar et al., 2020a; Qiu et al., 2021) without model retraining compromises model's accuracies on clean inputs (Sarkar et al., 2020b). Heuristic trigger search in image space (Udeshi et al., 2022; Xiang et al., 2022) does not scale to complex triggers or large image sizes.

To address the challenging task, we resort to the strong reconstruction power of modern image generation models. Intuitively, it can assist us to detect possible triggers and reconstruct the original clean image when the triggers are masked. We propose a novel method called *Blind Defense with Masked AutoEncoder* (BDMAE). Masked Autoencoders (He et al., 2022) are scalable self-supervised learners. They randomly mask patches from the input image and reconstruct the missing parts. Each patch corresponds to one of $14 \times 14$ tokens. Even using a high masking ratio (*e.g.*, 75%), the semantic contents can still be recovered. We can therefore search triggers efficiently in the token space. It enables us to generalize to complex trigger patterns or large image sizes.

Our method belongs to test-time image purification that incapacitates possible triggers while keeping semantic contents intact. We seek trigger scores that measure how likely each image patch contains triggers. High score regions are then removed and restored with MAE. The whole framework includes three main stages. First, we randomly generate MAE masks, and calculate two types of trigger scores based on image structural similarity and label prediction consistency between test images and MAE restorations, respectively. Then, we use the topology of triggers to refine both scores. The trigger scores help to generate topology-aware MAE masks that cover trigger regions more precisely, and the corresponding MAE restorations in turn help to refine trigger scores. Finally, we fuse multiple MAE restorations from adaptive trigger score thresholding into one purified image, and use that image for label prediction. Our approach is blind to the network architecture, trigger patterns or image benignity. It does not require additional training images for a particular test-time defense task. Extensive results demonstrate that BDMAE effectively purifies backdoored images without compromising clean images. BDMAE is generalizable to diverse trigger sizes and patterns.

Our main contributions are summarized as follows:

1. We address the practical task of blind backdoor defense at test time and for black-box models. Despite some general techniques for simple attacks, this challenging task has not been formally and systematically studied.
2. We propose to leverage generative models to assist backdoor defense. It may open a door to design general defense methods under limited data using abundant public foundation models.
3. A novel framework of Blind Defense with Masked Autoencoders (BDMAE) is devised to detect possible triggers and restore images on the fly. Three key stages are delicately designed to generalize to different defense tasks without tuning hyper-parameters.
4. We evaluate our method on four benchmarks, Cifar10 (Krizhevsky et al., 2009), GTSRB (Stallkamp et al., 2012), ImageNet (Deng et al., 2009) and VGGFace2 (Cao et al., 2018). Regardless of model architectures, image sizes or trigger patterns, our method obtains superior accuracies on both backdoored and clean images.

## 2 RELATED WORKS

**Backdoor attack.** BadNets (Gu et al., 2019) is the earliest work on backdoor attack. It attaches a checkerboard trigger to images and associates them with specific target labels. Many different trigger patterns are used in later works (Nguyen & Tran, 2020; Turner et al., 2019; Wenger et al., 2021).

These triggers are visible local patches in the images. Visible global triggers are used in (Chen et al., 2017; Barni et al., 2019). To make the attack stealthy, invisible patterns (Li et al., 2021c; Zhong et al., 2022; Zhao et al., 2022) and attacking strategies based on reflection phenomenon (Liu et al., 2020), image quantization and dithering (Wang et al., 2022c), style transfer (Cheng et al., 2021) and elastic image warping (Nguyen & Tran, 2021) are proposed. Although these stealthy attacks are less perceptible to humans, they are vulnerable to noise perturbations or image transformations. To make it hard for defenders to reconstruct triggers, sample-specific backdoor attacks (Li et al., 2021c; Nguyen & Tran, 2020) are proposed. This paper focuses on the visible triggers of local patches. The triggers can be either shared by samples or sample-specific.

**Backdoor defense.** Backdoor defense aims to mitigate backdoor behaviors. The training-stage defenses attempt to design robust training mechanism via decoupling training process (Huang et al., 2022), introducing multiple gradient descent mechanism (Li et al., 2021a) or modifying linearity of trained models (Wang et al., 2022b). However, intruding the training stage is often infeasible. Model reconstruction defenses mitigate backdoor behaviors by pruning bad neurons or retraining models using clean labeled data (Liu et al., 2018; Wang et al., 2019; Zeng et al., 2021). A recent work shows that backdoor behaviors could be cleansed by distillation on unlabeled data or even out-of-distribution data (Pang et al., 2023). Februus (Doan et al., 2020) is a test-time defense method. It detects triggers with GradCAM (Selvaraju et al., 2017), and feeds purified images to the model.

Recently, black-box backdoor models have drawn increasing attention (Chen et al., 2019; Dong et al., 2021; Guo et al., 2022; Zhang et al., 2021). In this setting, model parameters are concealed for data privacy or intellectual property. These works focus on identifying backdoored models, and usually reject predictions for such situations. Differently, we handle the task of blind backdoor defense at test time, aiming to obtain true label of every test image on the fly, with only access to the hard-label predictions. Test-time image transformation (Gao et al., 2019; Sarkar et al., 2020a; Qiu et al., 2021) and heuristic trigger search in image space (Udeshi et al., 2022) do not work well.

**Masked AutoEncoder.** Masked AutoEncoders (MAE) (He et al., 2022) are scalable self-supervised learners based on Vision Transformer (Dosovitskiy et al., 2021). It masks random patches of the input image, and restore the missing pixels. MAE has been used in many vision tasks (Bachmann et al., 2022; Pang et al., 2022; Tong et al., 2022; Xie et al., 2022; Chen et al., 2022; Li et al., 2022). Motivated by the powerful and robust data generation ability, for the first time we leverage MAE to detect triggers and restore images.

## 3 MOTIVATION AND INTUITION

Blind backdoor defense at test time aims to obtain correct label prediction for test images on-the-fly regardless the benignity of images and models. To solve this, test-time image purification is a viable solution that incapacitates backdoor triggers within images while keeping semantic contents intact. Some early works apply a global image transformation like blurring or shrinking (Sarkar et al., 2020a; Qiu et al., 2021; Li et al., 2021b). However, there is often a trade-off in selecting the strength. A stronger transformation is more likely to incapacitate the trigger but at a higher risk of ruining the semantic information. Recently, diffusion model based image purification methods (Nie et al., 2022; Wang et al., 2022a) leverage pretrained diffusion models to restore the content, but they highly reply on the image generation quality. When the test data distribution is different from the pretrained data distribution (*e.g.*, different image resolutions), the generated images may appear overall similar to the original test images but still different in the details. This makes it hard for the classifier to predict true labels.

Our motivation is to locate possible triggers and restore the missing contents simultaneously. The clean regions are kept intact. With this, the model predictions on clean images or clean models are minimally affected. Searching triggers in images can be challenging considering the diversity of trigger patterns and image sizes. Fortunately, with the help of pretrained Masked AutoEncoders (MAE), we can instead search triggers in the token space and use MAE to restore missing parts.

Compared with previous works, ours is fundamentally different in that
- We care about accuracies on both clean images and backdoored images, unlike other defense methods that only filter out backdoored images and refuse to make predictions on them.
- We leverage pretrained MAE models mainly to assist trigger search, unlike diffusion-based methods that leverage pretrained generative models to hallucinate the entire image contents.

Figure 2: Framework of our method. For a test image (may be backdoored), we generate the trigger score and refine it by considering the topology of triggers. The purified image obtained from adaptive restoration is used for making prediction.

## 4 METHODOLOGY

### 4.1 PROBLEM FORMULATION

We first formulate the backdoor attack and defense problems, then detail the proposed method of *Blind Defense with Masked AutoEncoder* (BDMAE) (Fig. 2). Our key idea is to detect possible triggers with the help of MAE.

**Backdoor attack.** Given a set of clean data $D = \{(\boldsymbol{x}, y)\}$, an adversary generates backdoored data $\tilde{D} = \{(\Phi(\boldsymbol{x}), \eta(y)) | (\boldsymbol{x}, y) \in D\}$, where $\Phi(\cdot)$ transforms a clean image into a backdoored image and $\eta(\cdot)$ transforms its true label into a target label. In this paper, we consider the popular formulation of $\Phi(\boldsymbol{x}) = (1 - \boldsymbol{b_x}) \odot \boldsymbol{x} + \boldsymbol{b_x} \odot \boldsymbol{\theta_x}$, where $\boldsymbol{b_x}$ is a binary mask, $\boldsymbol{\theta_x}$ is the backdoor trigger, $\odot$ denotes the Hadamard product (Dong et al., 2021; Hu et al., 2022; Zheng et al., 2021). $\eta(y)$ maps all true labels to one predefined target label. The mask and trigger may not be the same for different images. While triggers can span over the entire image, we only focus on local triggers that occupy a small area of the image. A prediction model $f$ is obtained by training on both clean data and backdoored data. In the situation without backdoor attack, $f$ is obtained from clean data only.

**Black-box test-time defense.** At test time, the suspicious model $f$ is provided as a black box and only its hard label predictions are accessible. The true label of each test image $\boldsymbol{x}$ needs to be recovered on the fly, without accessing additional data. To realize this, we seek a purified version $\rho(\boldsymbol{x})$ such that $f(\rho(\boldsymbol{x}))$ generates the correct label prediction. The test process is blind to the model or images, meaning that there is no information on whether $f$ is backdoored and whether $\boldsymbol{x}$ contains triggers. The goal is to achieve high classification accuracies on both clean and backdoored images.

### 4.2 TRIGGER SCORE GENERATION

For clarity, we assume that $f$ is backdoored and the test image $\boldsymbol{x}$ contains triggers. Our method can directly apply to clean models or clean images (*c.r.* Sec.4.5). Let $\hat{y} = f(\boldsymbol{x})$ be its original label prediction. To infer the trigger mask, one can repeatedly block some particular parts of the image and observe how model predictions change (Udeshi et al., 2022). However, the search space is huge for a common image size. Even worse, when the trigger is complex (*e.g.*, of irregular shape), the model may still predict the target label when some parts of the trigger remain in the image. These issues make the naïve trigger search method infeasible in practice.

We overcome the above-mentioned issues by leveraging the generic Masked AutoEncoders (MAE) (He et al., 2022). In MAE, each of the $14 \times 14$ tokens corresponds to a square patch of the image. MAE can recover the image content even when 75% tokens are masked out. This brings two benefits: 1) we can safely use a high masking ratio to remove triggers without changing the semantic label; and 2) since triggers are irrelevant to the content, they will unlikely present in the MAE restorations. To locate possible triggers, there are two complementary approaches: the **image-based** method that compares the *structural similarity* between the original image and MAE restorations, and the **label-based** method that compares the *consistency of label predictions* on the original image and MAE restorations.

We use both approaches to obtain an image-based trigger score matrix $S^{(i)} \in [0, 1]^{14 \times 14}$ and a label-based trigger score matrix $S^{(l)} \in [0, 1]^{14 \times 14}$. Each element of $S^{(i)}$ or $S^{(l)}$ thus implies how likely the corresponding image patch contains backdoor triggers. Compared with searching in the image space of size $H \times W$, searching in the token space of size $14 \times 14$ is much more efficient.

Before going to the method, we first describe how to restore $x$ given a pre-trained MAE $G$ and a token mask $m \in \{0,1\}^{14 \times 14}$. Define a function $\texttt{resize}(z; h, w)$ that resizes a tensor $z$ to size $h \times w$ by interpolation. As shown in Eq. 1, $x$ is first resized to $224 \times 224$ requested by MAE. Then we use $G$ to reconstruct the image based on $m$, and resize it back to $H \times W$. The additional steps aim to remove interpolation errors in the unmasked regions from the restoration $\tilde{x}$.

$$\bar{x} = \texttt{resize}\big(G(\texttt{resize}(x; 224, 224); m); H, W\big)$$
$$\tilde{m} = \texttt{resize}(m; H, W)$$
$$\tilde{x} = x \odot (1 - \tilde{m}) + \bar{x} \odot \tilde{m} \tag{1}$$
$$\tilde{G}(x, m) \triangleq (\tilde{x}, \tilde{m})$$

Now we describe how to obtain trigger scores $S^{(i)}$ and $S^{(l)}$ from MAE restorations. Let $\hat{y} = f(x)$ be its original hard-label prediction. We repeat the following procedure for $N_o$ times indexed by $o$. For each iteration, $N_i$ random token masks $\{m_{o,i} \in \{0,1\}^{14 \times 14}\}$ are sampled using a default masking ratio of 75%. The corresponding MAE reconstructions $\{\tilde{x}_{o,i}\}$ and masks $\{\tilde{m}_{o,i}\}$ are extracted from $\tilde{G}(x, m_{o,i})$ based on Eq. 1. Their hard-label predictions are $\{\hat{y}_{o,i} = f(\tilde{x}_{o,i})\}$.

**Image-based score** $S^{(i)}$. We fuse $N_i$ restorations into one image $\tilde{x}_o$ by:

$$\tilde{x}_o = \mathcal{F}\big(\{\tilde{x}_{o,i}\}, \{\tilde{m}_{o,i}\}\big) = \sum_i (\tilde{x}_{o,i} \odot \tilde{m}_{o,i}) \oslash \sum_i (\tilde{m}_{o,i}) \tag{2}$$

where $\odot$ and $\oslash$ are element-wise product and division. In Eq. 2, only image patches from MAE restorations are kept while other patches from the original image are discarded. The motivation is that triggers may not always be fully masked out, but we do not want them to appear in $\tilde{x}_o$. We manipulate the sampling of $\{m_{o,i}\}$ to guarantee that every image patch can be restored with Eq. 2.

The image-based score is defined as $S^{(i)} = \sum_o [1 - \texttt{resize}(\text{SSIM}(x, \tilde{x}_o); 14, 14)]/N_o$, averaged over $N_o$ repeated procedures. Here we use Structural Similarity Index Measure (SSIM) (Wang et al., 2004) to calculate the similarity between $\tilde{x}_o$ and $x$, where the SSIM score lies between $-1$ and $1$. As triggers are irrelevant to contents and unlikely present in $\tilde{x}_o$, SSIM scores in the trigger region will be low. In contrast, the clean regions will be well restored, leading to high SSIM scores.

**Label-based score** $S^{(l)}$. We average over token masks that lead to different label predictions. Formally, the label-based score is defined as $S^{(l)} = \sum_{o,i} [m_{o,i} \times (1 - \mathbb{I}[\hat{y} = \hat{y}_{o,i}])]/(N_o N_i)$, where $\mathbb{I}[\cdot]$ is the indicator function. The inconsistency in label predictions usually implies that triggers have been removed by the masks.

The two types of trigger scores are complementary to each other. $S^{(i)}$ favors large and complex triggers, while $S^{(l)}$ favors small triggers. Using both together can adapt to diverse trigger patterns.

## 4.3 TOPOLOGY-AWARE SCORE REFINEMENT

The trigger scores $S^{(i)}$ and $S^{(l)}$ obtained previously have high values for trigger regions. Nevertheless, they are still very noisy. The difference between scores of trigger regions and clean regions is also small, making it hard to determine a universal threshold for filtering out trigger regions.

We utilize the topology of triggers to refine trigger scores. Note that backdoor triggers are commonly continuous patterns (Hu et al., 2022). The obtained trigger scores indicate possible positions of triggers among the image. With the information, we can generate topology-aware MAE masks $\{m_r \in \{0,1\}^{14 \times 14}\}$ that cover trigger regions more precisely than uniformly sampled ones. This in turn guides us to enhance the difference between score values of clean regions and trigger regions. One issue is that if we apply refinement for all tokens, we may accidentally increase the score values of clean regions in the situation of clean images or clean models. To avoid this, we only focus on the top $L$ tokens that likely contain triggers, with $L = \sum_{r,c} \mathbb{I}[S_{r,c}^{(i)} \geq 0.2]$ or $L = \sum_{r,c} S_{r,c}^{(l)}$. Equivalently, a meta mask $m_{\text{rf}} \in \{0,1\}^{14 \times 14}$ can be defined, whose element is 1 only if the corresponding token belongs to the top $L$ tokens. $m_{\text{rf}}$ thus indicates the regions to be refined.

We use the same procedure to generate topology-aware MAE mask $m_r$ for both types of trigger scores. The main idea is to sequentially select tokens that have higher trigger scores or are adjacent to already selected tokens. For clarity, let $S^{(*)}$ denote either $S^{(i)}$ or $S^{(l)}$. We initialize $\mathcal{T} = \{t_0\}$ with

Table 1: Comparison with diffusion model based image purification method (500 test images).

| | | Cifar10 | | | GTSRB | | | VGGFace2 | | | ImageNet10 | | | ImageNet50 | | | ImageNet100 | | |
|---|---|---|---|---|---|---|---|---|---|---|---|---|---|---|---|---|---|---|---|
| | | CA | BA | ASR | CA | BA | ASR | CA | BA | ASR | CA | BA | ASR | CA | BA | ASR | CA | BA | ASR |
| Before Defense | | 94.0 | 1.0 | 99.0 | 99.7 | 1.1 | 98.9 | 96.7 | 0.0 | 100. | 88.6 | 9.4 | 89.4 | 84.4 | 0.7 | 99.2 | 81.7 | 0.3 | 99.6 |
| DiffPure | DDPM | 74.5 | 64.5 | 16.2 | 74.2 | 41.0 | 44.8 | 51.9 | 32.9 | 34.4 | 71.1 | 66.5 | 5.5 | 57.4 | 52.7 | 0.9 | 51.8 | 53.9 | 0.6 |
| | SDE | 75.5 | 63.9 | 15.6 | 71.7 | 42.8 | 44.4 | 52.8 | 33.6 | 35.5 | 73.2 | 67.6 | 5.1 | 54.1 | 57.2 | 1.1 | 52.8 | 53.8 | 0.7 |
| Ours | Base | 92.9 | 90.3 | 0.9 | 99.7 | 95.5 | 0.7 | 93.7 | 92.6 | 0.9 | 79.6 | 79.3 | 3.0 | 60.7 | 68.8 | 0.5 | 57.4 | 70.4 | 0.4 |
| | Large | 93.3 | 90.3 | 0.7 | 99.7 | 96.0 | 1.1 | 94.3 | 91.9 | 1.9 | 84.1 | 81.1 | 2.8 | 71.3 | 75.2 | 0.6 | 65.4 | 76.2 | 0.4 |

Table 2: Comparison with other image purification methods. ($\diamond$: white-box; others: black-box.)

| | | Cifar10 | | | GTSRB | | | VGGFace2 | | | ImageNet10 | | | ImageNet50 | | | ImageNet100 | | |
|---|---|---|---|---|---|---|---|---|---|---|---|---|---|---|---|---|---|---|---|
| | | CA | BA | ASR | CA | BA | ASR | CA | BA | ASR | CA | BA | ASR | CA | BA | ASR | CA | BA | ASR |
| Before Defense | | 93.3 | 0.9 | 99.0 | 98.5 | 1.4 | 98.6 | 95.5 | 0.0 | 100. | 89.5 | 9.7 | 89.2 | 84.0 | 0.5 | 99.4 | 82.3 | 0.2 | 99.8 |
| Februus$^\diamond$ | XGradCAM | 91.6 | 87.0 | 7.0 | 65.4 | 50.9 | 38.0 | 65.5 | 89.5 | 5.8 | – | – | – | – | – | – | – | – | – |
| | GradCAM++ | 80.0 | 91.0 | 2.3 | 59.1 | 73.9 | 14.6 | 63.1 | 89.4 | 5.9 | – | – | – | – | – | – | – | – | – |
| PatchCleanser | Vanilla | 89.9 | 43.9 | 55.0 | 95.0 | 10.0 | 89.7 | 93.0 | 43.0 | 56.9 | 84.5 | 58.0 | 37.1 | 79.6 | 45.7 | 49.4 | 78.9 | 43.4 | 52.3 |
| | Variant | 57.6 | 86.1 | 1.9 | 13.3 | 80.8 | 1.5 | 50.7 | 94.7 | 0.0 | 62.0 | 80.8 | 4.1 | 54.1 | 79.3 | 0.4 | 52.1 | 78.0 | 0.1 |
| Blur | Weak | 91.5 | 14.0 | 84.9 | 98.4 | 3.9 | 96.0 | 95.5 | 0.1 | 100. | 88.4 | 14.4 | 83.9 | 83.3 | 4.9 | 94.3 | 81.2 | 3.2 | 96.1 |
| | Strong | 63.6 | 60.0 | 6.4 | 97.7 | 94.9 | 1.8 | 95.2 | 10.4 | 89.4 | 84.8 | 34.2 | 60.9 | 79.2 | 49.1 | 39.3 | 76.0 | 51.6 | 33.1 |
| ShrinkPad | Weak | 90.7 | 50.3 | 45.0 | 97.5 | 33.3 | 65.0 | 93.8 | 35.5 | 62.5 | 88.4 | 43.0 | 52.1 | 82.0 | 39.7 | 51.1 | 80.0 | 42.3 | 46.0 |
| | Strong | 86.7 | 36.7 | 57.9 | 92.8 | 23.5 | 72.3 | 88.3 | 54.4 | 38.3 | 86.7 | 56.7 | 36.0 | 79.4 | 55.1 | 29.8 | 77.2 | 58.6 | 22.5 |
| Ours | Base | 92.5 | 90.8 | 0.9 | 98.2 | 95.3 | 0.9 | 91.3 | 92.0 | 1.6 | 79.9 | 81.1 | 4.8 | 61.7 | 70.1 | 0.8 | 59.0 | 67.9 | 0.4 |
| | Large | 92.7 | 91.1 | 0.8 | 98.4 | 96.0 | 0.9 | 92.9 | 91.8 | 2.2 | 83.9 | 83.7 | 3.9 | 72.6 | 76.1 | 0.6 | 69.5 | 73.9 | 0.3 |

token $t_0 = \arg\max_{t_k} S^{(*)}[t_k]$. Then we repeatedly add token $t_i = \arg\max_{t_k}(S^{(*)}[t_k] + 0.5\mathbb{I}[t_k \in \text{Adj}(\mathcal{T})]) \cdot \sigma_k$ to $\mathcal{T}$, where $\text{Adj}(\mathcal{T})$ includes all 4-nearest neighbors of tokens in $\mathcal{T}$ and $\sigma_k \sim U(0,1)$ is a random variable. This step achieves a balance between random exploration and topology-aware exploitation. The process continues until $|\mathcal{T}| = L/2$. The final $\mathcal{T}$ can be converted into an MAE mask $\boldsymbol{m}_r$, with its complementary part $\bar{\boldsymbol{m}}_r = \boldsymbol{m}_{\text{rf}} - \boldsymbol{m}_r$.

To refine the trigger score, we obtain the hard-label prediction $\hat{y}_r$ of MAE restoration based on $\boldsymbol{m}_r$. If $\hat{y}_r \neq \hat{y}$, we increase the score values of $S^{(*)}$ by a constant $\beta_0$ for tokens masked by $\boldsymbol{m}_r$ and $-\beta_0$ for other tokens; otherwise, we modify $S^{(*)}$ in an opposite way. Mathematically, $S^{(*)} \leftarrow S^{(*)} + (1 - 2\mathbb{I}[\hat{y} = \hat{y}_r]) \times \beta_0 \times (\boldsymbol{m}_r - \bar{\boldsymbol{m}}_r)$. Since $\|\boldsymbol{m}_r\|_0 = \|\bar{\boldsymbol{m}}_r\|_0 = L/2$, the average value of $S^{(*)}$ remains unchanged, while the contrast between trigger region and clean region are enhanced.

### 4.4 ADAPTIVE IMAGE RESTORATION

The combined trigger score used for label prediction is simply calculated as $S = (S^{(i)} + S^{(l)})/2$. One can convert $S$ into a binary mask based on some predefined threshold, and make prediction on the corresponding MAE restoration. However, the optimal threshold varies across different attack settings considering the diversity of image resolutions, backdoor attack methods, and trigger sizes.

We propose an adaptive image restoration mechanism to adapt to different attacks and datasets automatically. The idea is to fuse restorations from $K$ adaptive thresholds, $\{\tau_1 \geq \tau_2 \geq \cdots \geq \tau_K\}$. If $\sum_{r,c} \mathbb{I}[S_{r,c} \geq \tau_K]/(14 \times 14) \leq 25\%$ is not satisfied, we repeatedly increase all thresholds by a small amount. The rationale is that trigger regions should not dominate the image. These decreasing thresholds lead to a nest structure. We obtain the corresponding MAE restorations $\{\tilde{\boldsymbol{x}}_{\tau_k}, \tilde{\boldsymbol{m}}_{\tau_k} = \tilde{G}(\boldsymbol{x}, \boldsymbol{m}_{\tau_k})\}$, where $\boldsymbol{m}_{\tau_k}[r, c] = \mathbb{I}[S[r, c] \geq \tau_k]$, and then fuse them into one purified image $\rho(\boldsymbol{x}) = \mathcal{F}(\{\tilde{\boldsymbol{x}}_{\tau_k}\}, \{\tilde{\boldsymbol{m}}_{\tau_k}\})$. The model prediction $f(\rho(\boldsymbol{x}))$ is used for final evaluation.

### 4.5 GENERALIZATION TO CLEAN IMAGES AND CLEAN MODELS

Until now, we assume that both $f$ and $\boldsymbol{x}$ are backdoored. In practice, we deal with blind defense, meaning that both models and images can be either backdoored or clean. Our method directly applies to any of these situations, thanks to the dedicated designs. The effectiveness on clean images has been validated by CA metric. For clean models, we include discussions in Appendix Sec. E.

Table 3: Comparison results on three challenging attacks, IAB, LC and Blended. (`VF2` short for `VGGFace2`, and `IN10` short for `ImageNet10`.)

| | | Cifar10–IAB | | | GTSRB–IAB | | | Cifar10–LC | | | GTSRB–LC | | | VF2–Blended | | | IN10–Blended | | |
|---|---|---|---|---|---|---|---|---|---|---|---|---|---|---|---|---|---|---|---|---|
| | | CA | BA | ASR | CA | BA | ASR | CA | BA | ASR | CA | BA | ASR | CA | BA | ASR | CA | BA | ASR |
| Before Defense | | 93.4 | 1.6 | 98.4 | 98.0 | 1.2 | 98.7 | 94.5 | 0.5 | 99.5 | 95.8 | 5.3 | 94.7 | 95.1 | 1.9 | 98.1 | 86.5 | 28.4 | 68.4 |
| Februus◇ | XGradCAM | 91.7 | 29.9 | 68.1 | 68.4 | 72.5 | 24.8 | 92.6 | 63.7 | 33.9 | 80.6 | 91.7 | 5.1 | 68.9 | 74.6 | 21.4 | – | – | – |
| | GradCAM++ | 77.9 | 55.8 | 35.9 | 49.8 | 84.1 | 12.2 | 83.3 | 85.7 | 10.4 | 72.5 | 91.7 | 5.1 | 66.4 | 72.5 | 23.6 | – | – | – |
| PatchCleanser | Vanilla | 88.6 | 25.6 | 73.8 | 84.2 | 13.9 | 83.0 | 90.3 | 0.4 | 99.6 | 87.8 | 0.1 | 99.9 | 92.8 | 41.7 | 58.2 | 79.3 | 56.1 | 38.2 |
| | Variant | 62.2 | 66.7 | 26.3 | 16.6 | 83.0 | 6.7 | 56.5 | 4.7 | 95.3 | 9.8 | 6.0 | 94.0 | 47.0 | 93.4 | 1.4 | 56.5 | 68.7 | 15.5 |
| Blur | Weak | 91.3 | 33.9 | 63.7 | 97.8 | 18.6 | 81.2 | 92.5 | 92.3 | 0.7 | 95.5 | 95.1 | 1.2 | 95.1 | 38.0 | 60.7 | 86.5 | 47.5 | 46.8 |
| | Strong | 63.0 | 53.1 | 8.4 | 96.8 | 47.7 | 50.7 | 56.8 | 56.1 | 2.5 | 93.7 | 93.6 | 0.7 | 94.9 | 94.7 | 0.3 | 82.6 | 73.6 | 12.8 |
| ShrinkPad | Weak | 91.2 | 64.5 | 28.8 | 97.1 | 43.7 | 55.1 | 92.4 | 1.4 | 98.6 | 93.6 | 5.9 | 94.1 | 93.5 | 88.6 | 5.1 | 86.3 | 83.3 | 4.8 |
| | Strong | 88.5 | 80.6 | 7.1 | 93.1 | 62.2 | 32.5 | 89.7 | 85.1 | 6.1 | 81.8 | 68.0 | 23.1 | 87.0 | 86.1 | 1.0 | 84.5 | 80.3 | 5.4 |
| Ours | Base | 93.0 | 81.8 | 10.5 | 97.8 | 76.2 | 21.4 | 93.7 | 94.1 | 0.4 | 93.9 | 93.6 | 2.3 | 90.7 | 91.8 | 1.0 | 73.4 | 68.0 | 17.3 |
| | Large | 93.1 | 80.0 | 13.0 | 98.0 | 70.6 | 27.4 | 93.9 | 94.3 | 0.4 | 94.8 | 93.5 | 2.4 | 92.1 | 92.2 | 0.9 | 81.2 | 73.1 | 14.3 |

## 5 EXPERIMENTS

### 5.1 EXPERIMENTAL SETUP

**Datasets.** We evaluate our method on the commonly used `Cifar10` (Krizhevsky et al., 2009), `GTSRB` (Stallkamp et al., 2012), `VGGFace2` (Cao et al., 2018), and three `ImageNet` (Deng et al., 2009) subsets, including `ImageNet10`, `ImageNet50` and `ImageNet100`.

**Backdoor attacks settings.** We use BadNet (Gu et al., 2019) with different triggers, Label-Consistent backdoor attack (LC) (Turner et al., 2019), Input-Aware dynamic Backdoor attack (IAB) (Nguyen & Tran, 2020), and Blended attack (Chen et al., 2017) to build backdoored models. For `Cifar10` and `GTSRB`, the backbone network is ResNet18 (He et al., 2016) from random initialization. We conduct 14 repeated experiments from random target labels or initializations for each attack setting. For `VGGFace2` and `ImageNet`, we use pretrained ResNet50 (He et al., 2016) and conduct 6 repeated experiments. The backdoor triggers include white/color patches, small image patches, and random curves.

**Method configurations.** We use the publicly available Masked Autoencoders (He et al., 2022) pretrained on ImageNet to assist blind defense. The `Base` variant has 12 encoder layers, and the `Large` variant has 24 encoder layers with an increased hidden size dimension. The same hyper-parameters are used for all experiments. The initial thresholds used in our work is $\{0.6, 0.55, 0.5, 0.45, 0.4\}$.

**Baseline methods.** We compare with several different lines of methods. **Blur** and **ShrinkPad** (Li et al., 2021b) are two simple purification methods based on test-time image transformation. More transformations are discussed in Appendix Sec. F.5. **PatchCleanser** (Xiang et al., 2022) is a certifiably robust defense method against adversarial patches via double-masking. **DiffPure** (Nie et al., 2022) uses diffusion models for adversarial purification. In addition to these black-box methods, we also compare with a white-box method **Februus** (Doan et al., 2020) that uses GradCAM (Selvaraju et al., 2017) to locate triggers and restores missing parts with GAN models (Goodfellow et al., 2014).

**Evaluation metrics** include the classification accuracy on clean images (CA) and backdoored images (BA), as well as attack success rate (ASR). Due to page limit, we only report results averaged over all backdoor triggers in the main text, and leave detailed results in Appendix Sec. G.

### 5.2 MAIN RESULTS

**Comparison with diffusion model based DiffPure.** Since the diffusion sampling process is extremely slow, we only report results on 500 test images in Tab. 1. Overall, DiffPure can partially purify backdoored images but is much less effective than ours. DDPM and SDE sampling strategies obtain comparable performances. The low CA of DiffPure may be due to its reverse generative process that alternates image content, *e.g.*, on `VGGFace2` where face recognition heavily relies on fine-grained attributes. Another observation is high ASR of DiffPure. This method is originally proposed for imperceptible adversarial perturbation, and the backdoor triggers are hard to be completely removed with diffusion sampling. More details and analysis are provided in Appendix Sec. F.7.

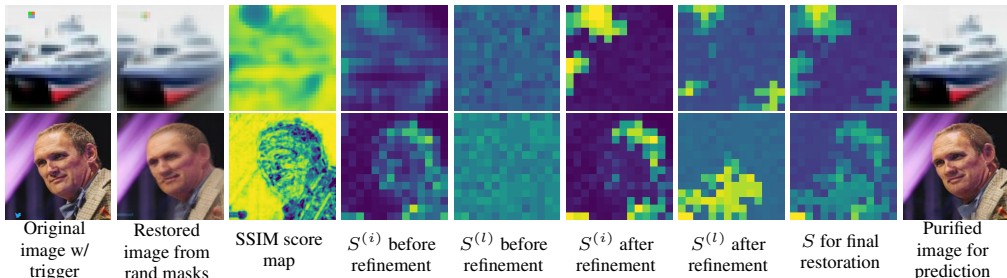

| Original image w/ trigger | Restored image from rand masks | SSIM score map | $S^{(i)}$ before refinement | $S^{(l)}$ before refinement | $S^{(i)}$ after refinement | $S^{(l)}$ after refinement | $S$ for final restoration | Purified image for prediction |

Figure 3: Sampled visualizations. Top: `Cifar10` with 2×2-color trigger. Bottom: `VGGFace2` with *twitter* trigger. All the scores are clipped to a range of [0,1], with yellow for high value.

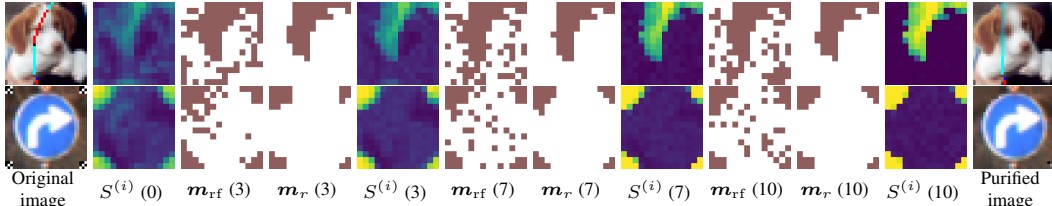

| Original image | $S^{(i)}$ (0) | $\boldsymbol{m}_{\mathrm{rf}}$ (3) | $\boldsymbol{m}_r$ (3) | $S^{(i)}$ (3) | $\boldsymbol{m}_{\mathrm{rf}}$ (7) | $\boldsymbol{m}_r$ (7) | $S^{(i)}$ (7) | $\boldsymbol{m}_{\mathrm{rf}}$ (10) | $\boldsymbol{m}_r$ (10) | $S^{(i)}$ (10) | Purified image |

Figure 4: Visualizations of topology-aware score refinement. Top: `Cifar10` with IAB. Bottom: `GTSRB` with LC. The numbers in brackets indicate steps of refinement.

**Comparison with other purification methods.** Table 2 lists results of other methods. For Februus, we substitute its original GradCAM with two recent improvements to work on complex backbone networks. The GAN models are released by the authors, yet unavailable for `ImageNet`. Februus successfully purifies backdoored images but it is not a black-box model. Its performance is sensitive to the CAM visualization. PatchCleanser uses two rounds of masking to locate the trigger patch. The inconsistency check step of vanilla method is frequently affected by noisy label predictions, leading to low BA and high ASR. We make a variant that can make decent predictions on backdoored images, but at a cost of much lower accuracies on clean images. The two simple test-time image transformations, Blur and ShrinkPad, both face a trade-off between CA and BA. Using a strong transformation is more likely to incapacitate backdoor triggers, but decreases clean accuracies.

Our method achieves high accuracies on both clean and backdoored images. For the two variants, using MAE-Large performs slightly better due to better restorations. Unlike Blur and ShrinkPad that apply global transformations, our method first locates possible triggers and then restore the trigger region only. Compared with Februus and PatchCleanser, our method leverages MAE model to better locate triggers. These two points are key to our excellent performance. We also want to highlight that Tab. 2 reports the aggregated results. Using different sizes of backdoor triggers may lead to different observations of these methods. Please refer to Appendix Sec. F for more discussions.

**Results on more challenging attacks.** In additional to the commonly used Backdoor attack with different triggers, we consider three more challenging attacks. IAB attack triggers are sample-specific irregular color curves or shapes, often split into a few fragments. LC attack triggers are checkerboards at the four corners. Blended attack triggers are invisible noise patches in the lower right corner. From Tab. 3, IAB and LC are more challenging for the comparison methods. The assumption of triggers covered by a small rectangle mask is invalid in PatchCleanser. The performances of comparison methods are rather inconsistent across different settings. Blur and ShrinkPad happen to be suitable choices for the invisible Blended attack. For all these challenging attack settings, our method obtains consistently high accuracies.

## 6    ANALYSIS

**Visualizations of defense process.** We plot images and scores in Fig. 3. Restored images from random masks have the same content as the original images, but are different in the trigger regions and some details. This is reflected in the SSIM score map. The two trigger scores are slightly higher in the trigger region, but very noisy. After refinement, high scores concentrate on the triggers, and

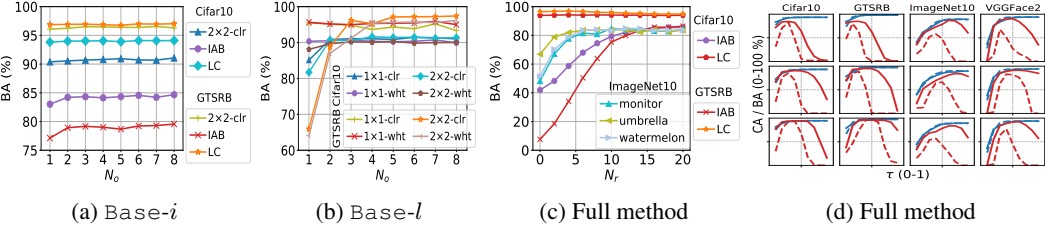

| (a) Base-$i$ | (b) Base-$l$ | (c) Full method | (d) Full method |

Figure 5: (a-c) Effects of $N_o$ and $N_r$. (d) Accuracies with fixed thresholds on backdoored/clean images, before (dashed) or after (solid) refinement. Refinement enlarges ranges of optimal thresholds.

scores of content regions are suppressed. $S$ is then used to generate the purified images. Compared with the original backdoored images, triggers are removed while the image contents are preserved. The purified images lead to correct label predictions.

**Effects of topology-aware refinement.** The topology-aware refinement is vital to the generalizability of our method. It exploits initialized scores, and generates topology-aware token masks to refine the scores. This is beneficial especially to complex triggers. In Fig. 4, the triggers are random curves and four distant checkerboards. Before refinement, the trigger regions have relatively high scores in $S^{(i)}$. But the contrast between trigger regions and clean regions are not significant. For each refinement, $m_r$ is sampled in a topology-aware manner to be continuous patches. $S^{(i)}$ is updated to have increased values for tokens masked by $m_r$ and reduced values for the rest. After 10 refinements, $S^{(i)}$ well reflects the trigger regions. It is worth mentioning that the refinement focuses on the triggers related to backdoor behaviors. Even though the blue line remains in the purified 'dog' image, the red line has been removed, thus it makes correct label prediction.

In Fig. 5c, we find that $N_r = 10$ is good enough for different triggers. One purpose of refinement is to increase contrast between scores of trigger regions and clean regions, so that the optimal threshold is easier to choose. In Fig. 5d, we randomly select three defense tasks for each dataset. Instead of fusing restorations from multiple thresholds, we choose a fixed threshold ranging from 0.1 to 0.9, and plot the accuracy curves. In each subplot, red/blue lines denote backdoored/clean images, dashed/solid lines denote before/after refinement. We can see that before refinement, the optimal thresholds have narrow ranges and vary across tasks. After refinement, they become wider. It is thus easy to set unified thresholds for different tasks.

**Sensitivity on hyper-parameters.** Our method mainly involves two critical hyper-parameters, the repeated times $N_o$ and the refinement times $N_r$. Throughout the experiments, we use $N_o = 5$ and $N_r = 10$. Figures 5a,5b plot the effects of $N_o$ in Base-$i$ and Base-$l$, respectively. For the image-based score $S^{(i)}$, the SSIM score map is similar for different MAE restorations. Thus averaging over 2 repeated results is good enough. For the label-based score $S^{(l)}$, averaging over many repeated results reduces the variance. $N_o = 5$ generally performs well for both scores.

**Discussion and Limitation.** Test-time backdoor defense has drawn increasing attention. It is a practical yet challenging task. Only model predictions on the single test image can be used, while the backdoor attack methods can be quite diverse. By leveraging pretrained MAE models, our method locates possible triggers inside images and restores the missing contents simultaneously. We demonstrate its effectiveness on backdoor triggers of different patterns and shapes. One limitation of our method is that it focuses on the most popular local triggers. Some particular attack methods use triggers that overlap the entire image. In that case, an additional step of image transformation can be applied before our method (Shi et al., 2023). We left that for an interesting future work.

## 7 CONCLUSION

In this paper, we study the novel yet practical task of blind backdoor defense at test time, in particular for black-box models. We devise a general framework of *Blind Defense with Masked AutoEncoder* (BDMAE) to detect triggers and restore images. Extensive experiments on four benchmarks under various backdoor attack settings verify its effectiveness and generalizability.

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

# A  DATASET DETAILS

`Cifar10` is a 10-class classification dataset (Krizhevsky et al., 2009) of size 32×32. There are 50,000 training images and 10,000 test images.

`GTSRB` (Stallkamp et al., 2012) consists of 43-class traffic signs images of size 32×32, split into 39,209 training images and 12,630 test images.

`VGGFace2` is a face recognition dataset (Cao et al., 2018). We use images from 170 randomly selected classes following (Doan et al., 2020), and resize them to 224×224. Face recognition is a critical real-world application where backdoor attack may exist.

`ImageNet10, ImageNet50 and ImageNet100` are three subsets of `ImageNet` (Deng et al., 2009), resized to 224×224. We created them by selecting the first 10 (50, 100) classes in alphabetical order. Each class has about 1,300 training images and 50 test images.

# B  IMPLEMENTATION DETAILS

## B.1  BACKDOOR ATTACK SETTINGS

We use BadNet attack (Gu et al., 2019) with different triggers, Label-Consistent backdoor attack (LC) (Turner et al., 2019), Input-Aware dynamic Backdoor attack (IAB) (Nguyen & Tran, 2020) and Blended attack (Chen et al., 2017) to build backdoored models.

The triggers of BadNet attack are chosen from $1 \times 1 \sim 3 \times 3$ white or color patches. In addition, we use several 15×15 icons as triggers for `VGGFace2` and `ImageNet`. These commonly seen object icons and social media icons are more natural in the real-world application. The triggers of LC attack are 3×3 checkerboards in the four images corners. The triggers of Blended attack is 15×15 random pixels. The triggers of IAB attack are random color curves or shapes. They are sample-specific, in that each image has its unique trigger pattern.

We randomly select 10% training data to create backdoored images. Then we train a model until it has a sufficiently high accuracy on clean images and attack success rate on backdoored images. The backbone network for `Cifar10` and `GTSRB` is ResNet18 (He et al., 2016) from random initialization. The backbone network for `VGGFace2` and `ImageNet` is pretrained ResNet50 (He et al., 2016). We also considered other backbone networks in Sec. F.3. For each setting of `Cifar10` and `GTSRB`, we report average results over 14 repeated experiments from different target labels or initializations. For the large `VGGFace2` and `ImageNet`, we reduce it to 6 repeat experiments.

## B.2  METHOD CONFIGURATIONS

We avoid tuning our method to some specific dataset or attack. Instead, we use the same set of hyper-parameters for all experiments. The motivation is that as only one test image is available in the task, it is unlikely to tune those hyper-parameters reliably. Specifically, the default masking ratio is 75%. $N_o = N_i = 5$ and $N_r = 10$. Even though, it is worthwhile mentioning that the image resolution and backdoor trigger patches can be highly diverse, better performances of our methods are expected with better tuned hyper-parameters.

We use two pretrained Masked Autoencoders (He et al., 2022) that are available from their official repository. The `Base` variant has 12 encoder layers, and the `Large` variant has 24 encoder layers with an increased hidden size dimension. For `Cifar10` and `GTSRB`, we up-sample each image to 224×224 first in order to fit MAE models. Afterwards, the MAE restorations are down-sampled back to the original image size.

The detailed procedures of our method can be found in Alg. 1 and Alg. 2.

## B.3  EXPERIMENT ENVIRONMENT

We experiment with Nvidia A5000 or A6000 GPUs using PyTorch 1.8. For the implementation, the MAE related code is adapted from its official repository.

---

**Algorithm 1** Trigger Score Generation

---

**Input:** Prediction model $f$, test image $\boldsymbol{x}$, generic MAE model $G$, repeated times $N_o$, $N_i$.
**Output:** Trigger scores $S^{(i)}$, $S^{(l)}$
 1: Get original hard-label prediction $\hat{y} = f(\boldsymbol{x})$
 2: **for** $o = 0$ **to** $N_o$ **do**
 3:     **for** $i = 0$ **to** $N_i$ **do**
 4:         Uniformly sample random token mask $\boldsymbol{m}_{o,i}$
 5:         Get MAE reconstruction $\tilde{\boldsymbol{x}}_{o,i}$ and the corresponding mask $\tilde{\boldsymbol{m}}_{o,i}$ from $\tilde{G}(\boldsymbol{x}, \boldsymbol{m}_{o,i})$
 6:         Get hard-label prediction $\hat{y}_{o,i} = f(\tilde{\boldsymbol{x}}_{o,i})$
 7:     **end for**
 8:     Fuse restorations into $\tilde{\boldsymbol{x}}_o = \mathcal{F}(\{\tilde{\boldsymbol{x}}_{o,i}\}, \{\tilde{\boldsymbol{m}}_{o,i}\})$
 9:     Calculate structural similarity $I_o = \text{SSIM}(\boldsymbol{x}, \tilde{\boldsymbol{x}}_o)$
10: **end for**
11: $S^{(i)} = \sum_o [1 - \texttt{resize}(I_o; 14, 14)]/N_o$
12: $S^{(l)} = \sum_{o,i} [\boldsymbol{m}_{o,i} \times (1 - \mathbb{I}[\hat{y} = \hat{y}_{o,i}])]/(N_o N_i)$

---

**Algorithm 2** Topology-aware Score Refinement

---

**Input:** Prediction model $f$, test image $\boldsymbol{x}$, generic MAE model $G$, refinement times $N_r$, initial trigger score $S^{(*)}$, mask $\boldsymbol{m}_{\text{rf}}$ for tokens to be refined, $\beta_0 = 0.05$.
**Output:** Refined trigger score $S^{(*)}$.
 1: Get original hard-label prediction $\hat{y} = f(\boldsymbol{x})$
 2: **for** $r = 0$ **to** $N_r$ **do**
 3:     Generate a topology-aware token mask $\boldsymbol{m}_r$
 4:     $\bar{\boldsymbol{m}}_r = \boldsymbol{m}_{\text{rf}} - \boldsymbol{m}_r$
 5:     Get MAE reconstruction $\tilde{\boldsymbol{x}}_r$ from $\tilde{G}(\boldsymbol{x}, \boldsymbol{m}_r)$
 6:     Get hard-label prediction $\hat{y}_r = f(\tilde{\boldsymbol{x}}_r)$
 7:     $\beta = (1 - 2\mathbb{I}[\hat{y} = \hat{y}_r]) \times \beta_0$
 8:     $S^{(*)} \leftarrow S^{(*)} + \beta \times (\boldsymbol{m}_r - \bar{\boldsymbol{m}}_r)$
 9: **end for**

---

## C   Comparison Methods

**Februus** (Doan et al., 2020) is a **white-box** defense method. It uses GradCAM (Selvaraju et al., 2017) visualization to locate image regions that are more relevant to the backdoor target label. It is highly likely that backdoor triggers are inside these regions. Then Februus removes those regions and uses a separately-trained GAN (Goodfellow et al., 2014) for image restoration. We use the GAN models provided by the authors and skip Februus experiments on `ImageNet` as the corresponding GAN model is unavailable.

GradCAM requires knowing detailed model architecture and parameters. In our experiments, we found that GradCAM does work well on models with deeper layers and more complex classifier heads. Therefore, we substitute it with two improved versions, XGradCAM (Fu et al., 2020) and GradCAM++ (Chattopadhay et al., 2018). Another critical issue with Februus is the layer for visualization and the threshold for selecting highly relevant regions. The choices significantly affects the performance on clean images and backdoor images. We first find the best layer for each model architecture, and the try all thresholds from $\{0.6, 0.7, 0.8\}$. The reported results are from thresholds with the best $ACC_c + ACC_b$ for each defense setting individually.

**PatchCleanser** is a certifiably robust defense against adversarial patches (Xiang et al., 2022). It assumes that the entire adversarial patches can be covered by a small rectangle mask, which may not be applicable to LC and IAB attacks in our task. The method performs two rounds of pixel masking on the image to neutralize the effect of the adversarial patch. If the first mask covers the adversarial patch, then moving the second mask will not change the predicted label. Otherwise, the second round masking may exhibit in-consistent label predictions.

We adapt the official PatchCleanser implementation to our backdoor defense framework. However, we found that the vanilla method may fail to locate backdoor triggers frequently. The reason is that the inconsistency check mentioned above could be affected by noisy label prediction that is neither real semantic label nor target label. It fails to return the disagreer prediction (Alg.1 Ln. 8-10 of the original paper), instead the majority prediction (*i.e.*, target label) is returned. This results in low BA and high ASR on backdoored images. We proposed a variant that returns the majority voting of all predictions for the in-consistent situations. It works much better to purify backdoored images, but at an expense of lower accuracies on clean images.

**Blur** is a simple test-time image transformation method used in (Li et al., 2021b). We implemented it with a $3\times3$ Gaussian kernel of standard deviation 0.5 for weak blurring and 1.0 for strong blurring.

**ShrinkPad** is proposed by (Li et al., 2021b) that first shrinks images and then randomly pads images to the original size. We adapt the authors' implementation to our framework. For small images from `Cifar10` and `GTSRB`, the weak and strong transformation use padding sizes of 4 and 8, respectively. For large images from `VGGFace2` and `ImageNet`, the padding sizes are 28 and 56, respectively.

**DiffPure** (Nie et al., 2022) uses pre-trained diffusion models to purify images. It first diffuses an image with a small amount of noise following a forward diffusion process, and then recover the clean image through a reverse generative process. Our implementation is based on their official repository. We use the public $256\times256$ unconditional diffusion model from OpenAI. Each test image is up-sampled to $256\times256$ for diffusion model and down-sampled back to the original image size. For the diffusion sampling strategy, we use both DDPM and SDE following DiffPure. We found that the diffusion restorations may not always stick to the original content. The issue is more severe for low-resolution images.

## D    REMARKS ON SSIM

Structural Similarity Index Measure (SSIM) (Wang et al., 2004) is used to measure the similarity between two images. Different from Mean-Squared-Error that measures pixel-wise absolution errors, SSIM considers the inter-dependencies among neighboring pixels. The SSIM index is calculated on two windows, $x$ and $y$, from a pair of images. Its definition is

$$\text{SSIM}(x, y) = \frac{(2\mu_x\mu_y + c_1)(2\sigma_{xy} + c_2)}{(\mu_x^2 + \mu_y^2 + c_1)(\sigma_x^2 + \sigma_y^2 + c_2)} \tag{A.3}$$

where $\mu_x$ and $\mu_y$ are mean values, $\sigma_x$ and $\sigma_y$ are variances, and $\sigma_{xy}$ is covariance. $c_1$ and $c_2$ are constants. $\text{SSIM}(x, y)$ lies between -1 and 1. 1 indicates perfect similarity, 0 indicates no similarity, and -1 indicates perfect anti-correlation. In our experiments, we observe that the minimum SSIM values are about -0.6$\sim$-0.2 depending on datasets, and the maximum values are close to 1.0.

The window size influences the SSIM values. Generally, a larger window averages over more pixels, thus the SSIM value is less extreme (*i.e.*, close to 0). We use the commonly used $11\times11$ Gaussian window, whose effective window size is about $5\times5$. On `Cifar10` and `GTSRB` of image size $32\times32$, due to their low resolution, a $11\times11$ window usually covers content regions. The original image and MAE restorations are similar, thus it is unlikely that the SSIM values will be extremely negatively. On `ImageNet` and `VGGFace2` of image size $224\times224$, differently, the window may include some background regions or image details. The difference between the original image and MAE restoration can be significantly large, leading to significantly negative SSIM values. Since our image-based trigger score is defined as $S^{(i)} = 1 - \text{SSIM}$, $S^{(i)}$ tends to be larger for `ImageNet` and `VGGFace2`. This is why the adaptive thresholds are necessary to achieve good performance on the two datasets.

## E    GENERALIZATION TO CLEAN IMAGES AND CLEAN MODELS

We highlight that our method is blind to the benignity of images or models. This relies on dedicated designs in different stages. The key to guarantee correct label prediction in the situation of clean images or clean models is not destroying semantic contents in clean regions. Since the final prediction is based on MAE restoration $\rho(\boldsymbol{x})$ from the final trigger score $S$, we should keep small score values of $S$ for those clean regions throughout the test-time defense process.

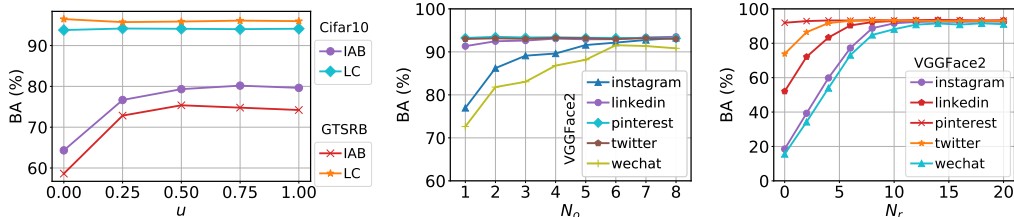

Figure A.1: Parameter analysis in the full method. Left: sampling parameter $u$ in topology-aware token mask generation. Center: repeated times $N_o$. Right: refinement times $N_r$.

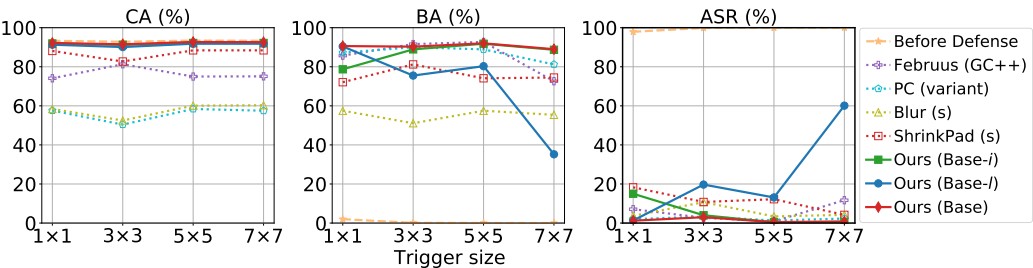

Figure A.2: Varying trigger (checker-board) sizes on `Cifar10`.

In the trigger score generation stage, if $x$ is a clean image no matter $f$ is backdoored or not, its MAE restorations should be similar to the original image. This implies that values of $S^{(i)}$ will be small. The values of $S^{(l)}$ will also be small as the label prediction is unlikely to change. If $f$ is a clean model and $x$ is a backdoored image, $S^{(l)}$ will still be small. Although $S^{(i)}$ has high values for trigger region, its impact is reduced when we average $S^{(l)}$ and $S^{(i)}$. In the topology-aware score refinement stage, only the top $L$ tokens are affected. By construction $L = \sum_{r,c} \mathbb{I}[S^{(i)}_{r,c} \geq 0.2]$ or $L = \sum_{r,c} S^{(l)}_{r,c}$, $L$ is generally small in the situation of clean images or clean models. In the adaptive image restoration stage, image regions with trigger scores greater than $\tau_K$ are generated with MAE. These regions are either trigger regions or some content-irrelevant regions. The rest clean content regions are kept intact. Therefore, the model can still make correct label prediction on the purified image $\rho(x)$.

For backdoored models on clean images, the CA in previous results has validated the effectiveness of our method. Figure A.3 shows different properties of trigger score $S$ between backdoored and clean images. $S$ of clean images has small values, thus the image restoration stage will not change the semantic content. For clean models, Table A.2 lists prediction accuracies for six different datasets. As can be seen, the accuracies on backdoored and clean images are minimally affected.

# F  ADDITIONAL ANALYSES

## F.1  TOPOLOGY-AWARE TOKEN MASK GENERATION

In the score refinement, we generate topology-aware token masks. We repeatedly choose $t_i = \arg\max_{t_k}(S^*[t_k] + u[\![t_k \in \text{Adj}(\mathcal{T})]\!]) \cdot \sigma_k$ with $u = 0.5$, where $\text{Adj}(\mathcal{T})$ includes all 4-nearest neighbors of tokens in $\mathcal{T}$ and $\sigma_k \sim U(0, 1)$ is a random variable. Here $u$ is the additional probability assigned to the neighboring tokens. When $u = 0$, the sampling procedure only select tokens with highest trigger scores. To see the effect of $u$, Fig. A.1 plots the results on four defense tasks with increasing $u$. For the challenging IAB attacks, the performances drops when not using topology-aware sampling (*i.e.*, $u = 0$). $u = 0.5$ obtains relatively good performances on the four tasks. Note that due to the existence of random variable $\sigma_k$, using $u = 1.0$ still leads to some randomness in the token selection.

Table A.1: Varying backbone network architectures.

| | | Cifar10 | | | | | | | | | VGGFace2 | | | | | | | | |
| | | ResNet18 | | | 6 Conv + 2 Dense | | | VGG16 | | | ResNet18 | | | ResNet50 | | | VGG16 | | |
| | | CA | BA | ASR | CA | BA | ASR | CA | BA | ASR | CA | BA | ASR | CA | BA | ASR | CA | BA | ASR |
|---|---|---|---|---|---|---|---|---|---|---|---|---|---|---|---|---|---|---|---|
| Before Defense | | 92.8 | 0.1 | 99.9 | 91.3 | 0.0 | 100. | 89.8 | 0.0 | 100. | 94.0 | 0.0 | 100. | 95.5 | 0.0 | 100. | 91.5 | 0.0 | 100. |
| Februus | XGradCAM | 91.0 | 83.9 | 11.0 | 86.2 | 90.0 | 2.7 | 87.3 | 45.7 | 50.0 | 46.3 | 93.6 | 0.2 | 65.5 | 89.5 | 5.8 | 81.3 | 75.6 | 17.7 |
| | GradCAM++ | 88.0 | 88.4 | 6.1 | 91.3 | 91.2 | 1.5 | 76.2 | 77.5 | 15.1 | 43.5 | 92.8 | 1.1 | 63.1 | 89.4 | 5.9 | 80.5 | 77.1 | 16.1 |
| PatchCleanser | Vanilla | 89.2 | 33.1 | 66.8 | 87.0 | 36.8 | 62.9 | 86.5 | 29.7 | 70.1 | 92.0 | 36.4 | 63.6 | 93.0 | 43.0 | 56.9 | 88.3 | 35.6 | 64.2 |
| | Variant | 50.4 | 90.0 | 3.3 | 52.6 | 88.2 | 2.8 | 47.8 | 86.6 | 2.8 | 45.0 | 93.1 | 0.0 | 50.7 | 94.7 | 0.0 | 42.8 | 90.3 | 0.1 |
| Blur | Weak | 90.5 | 90.0 | 2.3 | 87.5 | 73.2 | 20.0 | 86.2 | 27.5 | 69.6 | 93.9 | 0.0 | 100. | 95.5 | 0.1 | 100. | 88.9 | 69.5 | 22.8 |
| | Strong | 52.5 | 51.1 | 10.7 | 55.0 | 53.7 | 7.3 | 54.7 | 53.6 | 6.8 | 93.7 | 14.2 | 85.4 | 95.2 | 10.4 | 89.4 | 88.3 | 78.8 | 11.3 |
| ShrinkPad | Weak | 86.8 | 24.5 | 74.5 | 83.4 | 23.5 | 75.8 | 82.1 | 1.6 | 98.3 | 91.8 | 12.1 | 87.3 | 93.8 | 35.5 | 62.5 | 88.2 | 5.4 | 93.8 |
| | Strong | 82.7 | 81.2 | 10.8 | 73.8 | 70.5 | 23.4 | 74.1 | 72.1 | 16.3 | 83.5 | 24.9 | 71.1 | 88.3 | 54.4 | 38.3 | 72.6 | 25.2 | 52.3 |
| Ours | Base | 91.5 | 90.3 | 3.0 | 89.9 | 90.0 | 1.3 | 88.5 | 87.8 | 2.5 | 87.9 | 88.1 | 4.2 | 91.3 | 92.0 | 1.6 | 83.7 | 84.5 | 5.2 |
| | Large | 91.7 | 91.1 | 2.1 | 90.3 | 90.2 | 1.2 | 88.8 | 88.3 | 2.2 | 90.5 | 88.0 | 4.6 | 92.9 | 91.8 | 2.2 | 85.5 | 85.3 | 4.5 |

Table A.2: Defense results on clean models.

| | | Cifar10 | | GTSRB | | VGGFace2 | | ImageNet10 | | ImageNet50 | | ImageNet100 | |
| | | CA | BA | CA | BA | CA | BA | CA | BA | CA | BA | CA | BA |
|---|---|---|---|---|---|---|---|---|---|---|---|---|---|
| Before Defense | | 93.8 | 93.7 | 98.7 | 98.6 | 95.7 | 95.7 | 89.8 | 88.8 | 84.2 | 83.6 | 82.7 | 82.2 |
| Ours | Base | 93.1 | 93.1 | 98.5 | 98.5 | 91.5 | 91.4 | 81.3 | 80.7 | 61.8 | 61.4 | 59.3 | 59.6 |
| | Large | 93.3 | 93.2 | 98.6 | 98.6 | 92.8 | 92.6 | 85.3 | 84.1 | 72.5 | 72.5 | 69.8 | 70.1 |

## F.2 Sensitivity on Hyper-parameters

Figure A.1 shows additional analysis of hyper-parameter sensitivity on `VGGFace2`. As can be seen, using larger repeated times $N_o$ and refinement times $N_r$ leads to higher accuracies. $N_o = 5$ and $N_r = 10$ are good enough, which is consistent with our observations on other datasets in Fig. 5.

## F.3 Generalization on network architecture

Our method is for black-box defense, thus is generalizable on network architectures. In Tab. A.1, we show results on `Cifar10` and `VGGFace2` with different backbone networks. Februus is a white-box method, thus it relies on the network architecture. On `Cifar10`, it performs well on the shallow Convolutional Neural Network originally used by the authors, but is less effective on ResNet18 and VGG16. On `VGGFace2`, its clean accuracies are relatively low. Compared with other black-box methods, our method achieves consistently better performances across different network architectures.

## F.4 Varying Trigger Size

The trigger size affects the difficulty to detect these triggers. In the BadNet work (Gu et al., 2019), the authors use 3×3-checkerboard as triggers. In Fig. A.2, we present defense results on `Cifar10` using various sizes of checkerboard triggers. The performances of comparison methods are discouraging. Our method maintains high accuracies on clean images. Our `Base`$-i$ is not working well on 1×1 trigger because it is hard to detect such a small trigger using image similarity. `Base`$-l$, on the contrary, works well on this small trigger using label consistency. As trigger size becomes larger, the performance of `Base`$-l$ drops because the trigger can not be removed completely through random masking. The full method `Base` combines the merits of both image similarity and label consistency, and works for all cases.

## F.5 Defense with More Test-Time Transformations

To defense against backdoor attack, test-time transformations have been used in some previous works (Gao et al., 2019; Sarkar et al., 2020a; Qiu et al., 2021). Since they are training free and can

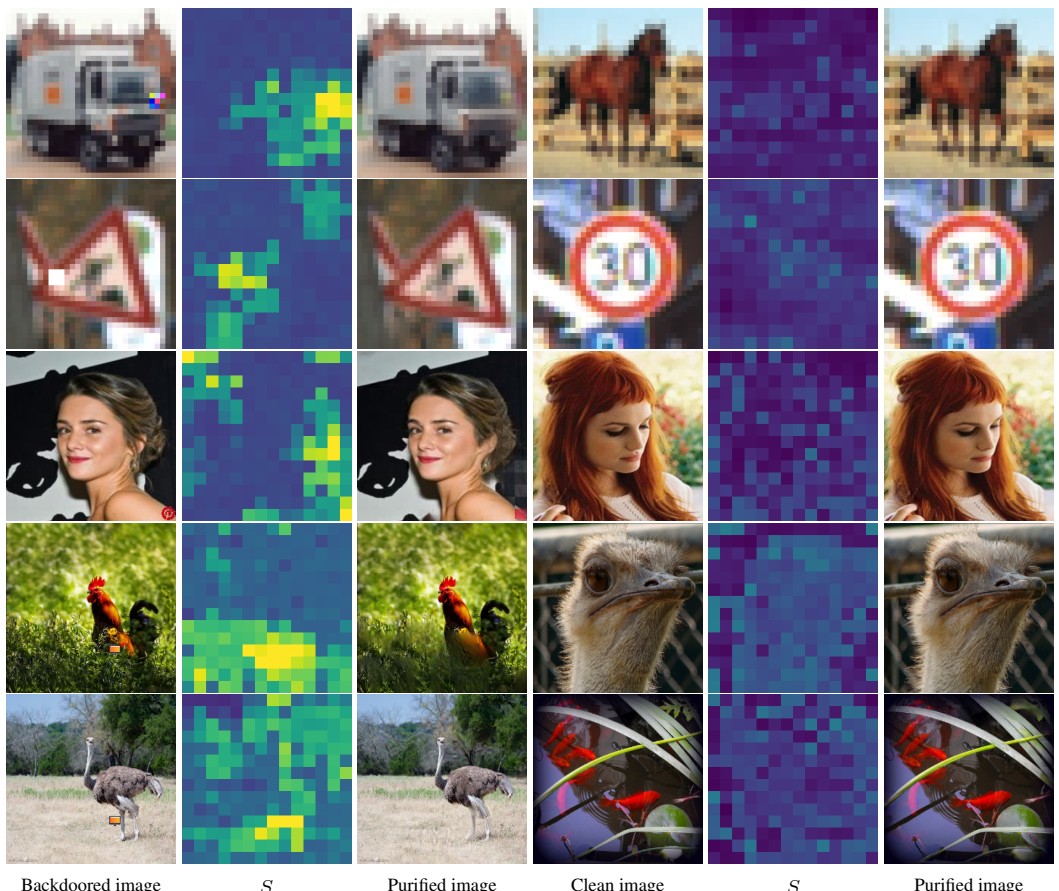

| Backdoored image | $S$ | Purified image | Clean image | $S$ | Purified image |

Figure A.3: Visualizations on backdoored / clean images.

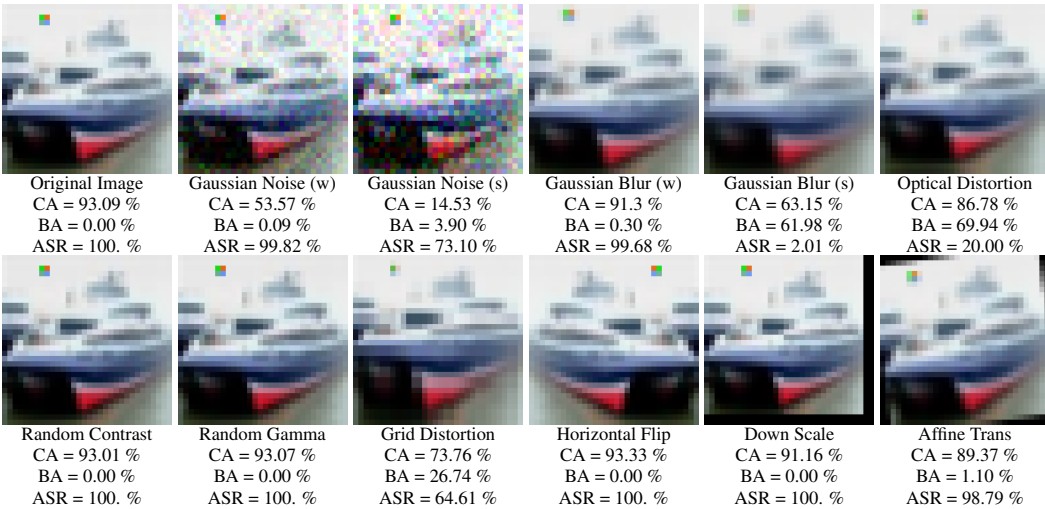

| Original Image | Gaussian Noise (w) | Gaussian Noise (s) | Gaussian Blur (w) | Gaussian Blur (s) | Optical Distortion |
| CA = 93.09 % | CA = 53.57 % | CA = 14.53 % | CA = 91.3 % | CA = 63.15 % | CA = 86.78 % |
| BA = 0.00 % | BA = 0.09 % | BA = 3.90 % | BA = 0.30 % | BA = 61.98 % | BA = 69.94 % |
| ASR = 100. % | ASR = 99.82 % | ASR = 73.10 % | ASR = 99.68 % | ASR = 2.01 % | ASR = 20.00 % |
| Random Contrast | Random Gamma | Grid Distortion | Horizontal Flip | Down Scale | Affine Trans |
| CA = 93.01 % | CA = 93.07 % | CA = 73.76 % | CA = 93.33 % | CA = 91.16 % | CA = 89.37 % |
| BA = 0.00 % | BA = 0.00 % | BA = 26.74 % | BA = 0.00 % | BA = 0.00 % | BA = 1.10 % |
| ASR = 100. % | ASR = 100. % | ASR = 64.61 % | ASR = 100. % | ASR = 100. % | ASR = 98.79 % |

Figure A.4: Defense results of applying test-time image transformations on `Cifar10` with 2×2-color trigger. The metrics shown are calculated on the entire test set.

be applied to our task, we briefly summarize these methods and remark on their limitation in our blind backdoor defense setting. **Supression** (Sarkar et al., 2020a) creates multiple fuzzed copies of backdoored images, and uses majority voting among fuzzed copies to recover label prediction. The fuzzed copies are obtained by adding random uniform noise or Gaussian noise to the original

image. However, the intensity of noise is critical. Weak noise would not remove the backdoor behaviour, while strong noise may destroy the semantic content. **DeepSeep** (Qiu et al., 2021) mitigates backdoor attacks using data augmentation. It first fine-tunes the infected model via clean samples with an image transformation policy, and then preprocesses inference samples with another image transformation policy. The image transformation functions include affine transformations, median filters, optical distortion, gamma compression, *etc*. The fine-tuning stage requires additional clean samples, which are unavailable in our setting. **STRIP** (Gao et al., 2019) superposes a test image with multiple other samples, and observes the entropy of predicted labels of these replicas. It aims to detect backdoored inputs, but could not locate the triggers nor recover the true label.

In Fig. A.4, we try different test-time image transformations on `Cifar10` with $2\times2$-color trigger. For each transformation, we calculate the CA on clean images, BA and ASR on backdoored images. As can be seen, some weak transformations, like Gaussian Noise (w), Gaussian Blur (w), Random Contrast/Gamma, Horizontal Flip and Down Scale, can not reduce ASR. While the rest strong transformations reduces ASR, they also compromise accuracies on clean images unacceptably. To maintain performance on clean images, the model needs to adapt to these image transformations, *e.g.*, through fine-tuning like DeepSeep does. Such requirement is infeasible in the blind backdoor defense, especially for black-box models.

### F.6 ADDITIONAL VISUALIZATION OF DEFENSE PROCESS

We present additional visualization of the defense process in Fig. A.5. The top six rows come from IAB (Nguyen & Tran, 2020) attack. IAB uses sample-specific triggers, *i.e.*, test images contain different triggers for one backdoored model. On `Cifar10`, the triggers are irregular curves. On `GTSRB`, the triggers are color patches. Due to the complexity of triggers, the heuristic search in image space using rectangle trigger blockers (Udeshi et al., 2022) may not work well. In our method, the refined trigger score $S$ successfully identifies the trigger in each test image. Triggers are removed in the purified images, leading to correct label predictions. On `VGGFace2` and `ImageNet10`, despite their larger image size, our method also manages to locate the tiny triggers and restore the clean images.

### F.7 ADDITIONAL VISUALIZATION OF DEFENSE RESULTS

Figure A.6 and Figure A.7 visualize the purified images of comparison defense methods. ShrinkPad and Blur apply global transformations on the images. They cannot remove the backdoor triggers, but sometimes can incapacitate the backdoor triggers through adding noises or distorting the trigger patterns. When the trigger patterns are large (*i.e.*, IAB in Figure A.7), a strong transformation would be required to reduce ASR. But this will also sacrifice clean accuracies.

DiffPure first adds a small amount noise to the backdoored images, and then uses a reserve generative process to recover the clean images. However, it frequently hallucinates image content. Looking at the last three columns of `VGGFace2`, DiffPure changes the facial expressions and facial features. These fine-grained attributes are critical to face recognition. For other datasets, DiffPure may not recover digits of `GTSRB` and the trigger patterns remain in the images. Although sometimes the trigger patterns are incapacitated. These visualization clearly shows the difference between our method and DiffPure, even though they both leverage large pretrained generative models. Ours only restores trigger-related regions, and keep other clean regions that contains important semantic details intact.

## G DETAILED RESULTS

We present the full results in Tables. A.3-A.8. Februus (Doan et al., 2020) uses GradCAM visualization to locate backdoor triggers. It relies on a threshold parameter to determine the backdoor removal regions. In the original paper, the authors use a held-out test set to determine this parameter for each dataset. Since we do not have such held-out test set in our blind backdoor defense task, we try with $\{0.6, 0.7, 0.8\}$, and report results with the parameter leading to best (CA+BA)/2 in the paper. The best parameter is selected for each attack setting individually. Februus is quite sensitive to this parameter. Generally, using a smaller parameter improves BA but reduces CA in Februus. The

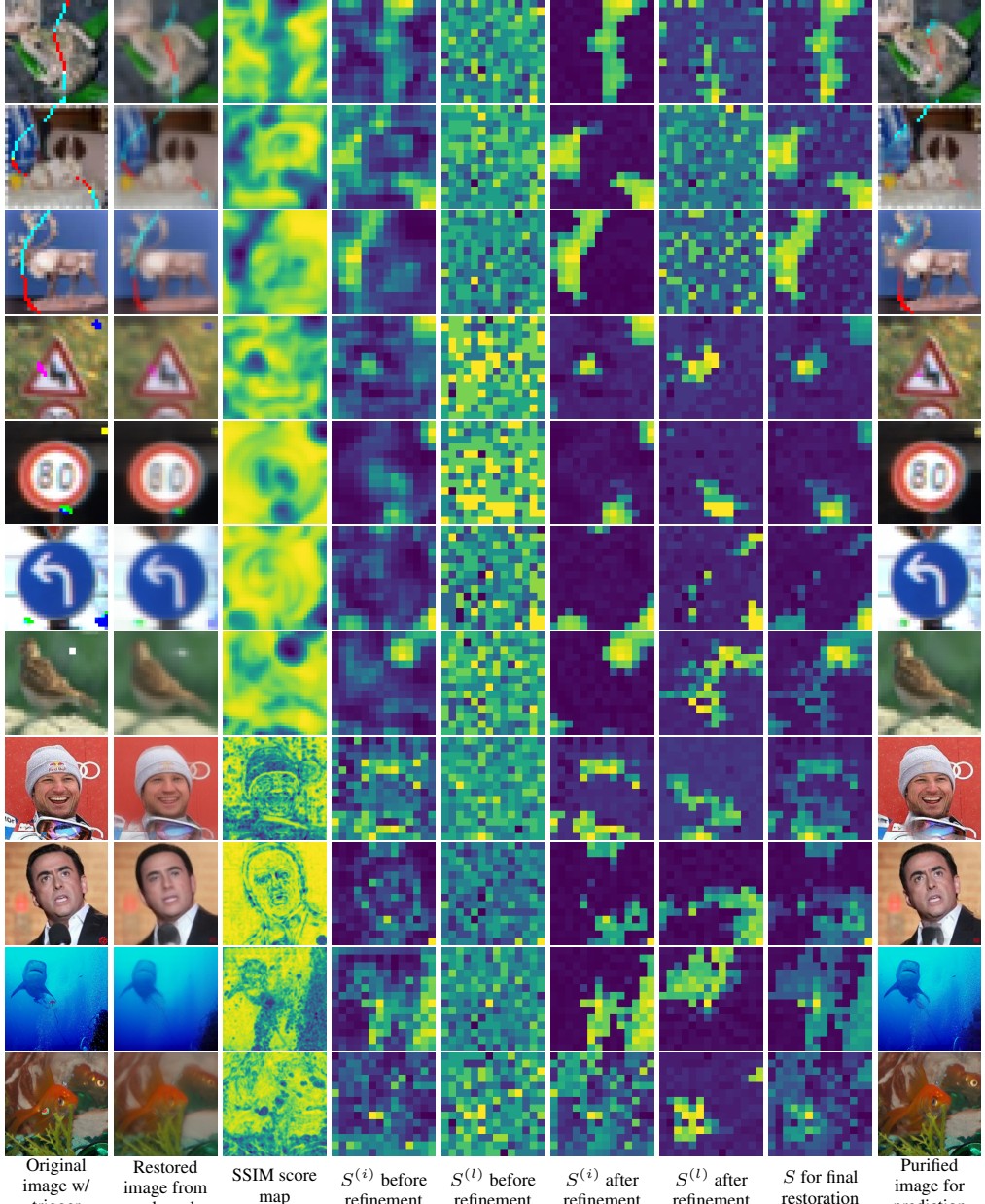

| Original image w/ trigger | Restored image from rand masks | SSIM score map | $S^{(i)}$ before refinement | $S^{(l)}$ before refinement | $S^{(i)}$ after refinement | $S^{(l)}$ after refinement | $S$ for final restoration | Purified image for prediction |

Figure A.5: Sampled visualizations of the defense process. All the scores are clipped to a range of [0,1], with yellow for high value. The top six rows are from IAB attack, and the rest are from BadNet attack.

best parameter varies across different defense tasks. Our method achieves a good balance between accuracies on clean images and backdoored images.

In the main text, we report aggregated results over different backdoor triggers. However, the defense performances can be quite different depending on the trigger sizes and patterns. Our method, using two complementary trigger scores, is designed to achieve decent accuracies in all situations.

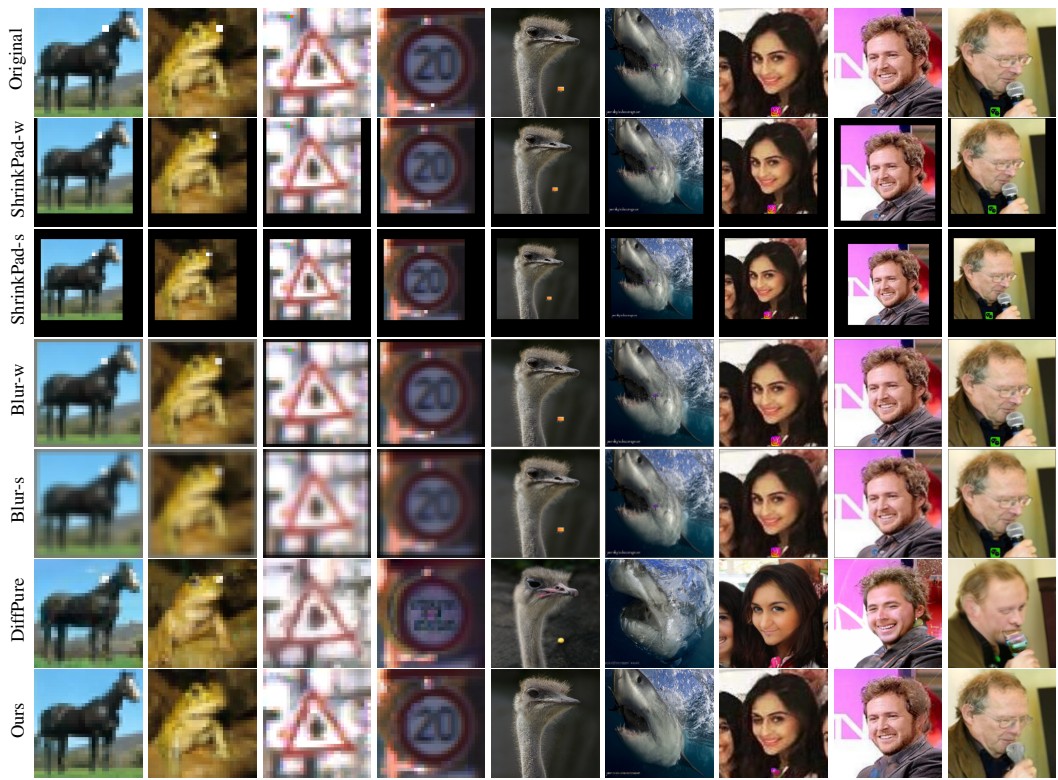

Figure A.6: Sampled visualizations of original images with triggers and images after defense (I).

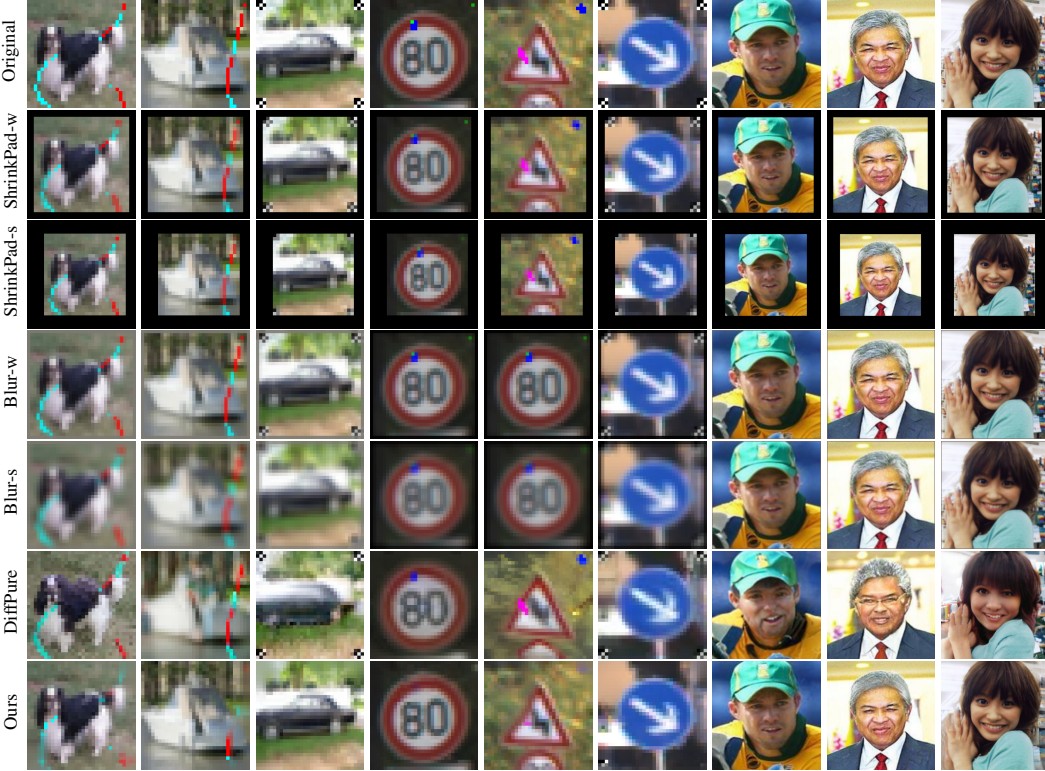

Figure A.7: Sampled visualizations of original images with triggers and images after defense (II).

Table A.3: Defense results on `Cifar10` using various sizes of color/white triggers.

| | | 1×1-color | | | 1×1-white | | | 2×2-color | | | 2×2-white | | | 3×3-color | | | 3×3-white | | |
|---|---|---|---|---|---|---|---|---|---|---|---|---|---|---|---|---|---|---|---|---|
| | | CA | BA | ASR | CA | BA | ASR | CA | BA | ASR | CA | BA | ASR | CA | BA | ASR | CA | BA | ASR |
| Before Defense | | 93.7 | 0.1 | 99.9 | 92.9 | 2.3 | 97.5 | 93.2 | 0.0 | 100. | 93.1 | 2.8 | 97.1 | 93.7 | 0.0 | 100. | 93.4 | 0.5 | 99.5 |
| Februus | XGradCAM (0.6) | 92.0 | 85.7 | 7.8 | 91.1 | 80.7 | 13.7 | 91.3 | 93.1 | 0.8 | 91.2 | 92.6 | 1.2 | 92.1 | 92.9 | 1.0 | 91.6 | 77.1 | 17.7 |
| | XGradCAM (0.7) | 92.8 | 78.5 | 16.2 | 91.9 | 56.1 | 40.4 | 92.2 | 87.3 | 7.0 | 92.1 | 85.0 | 9.5 | 92.8 | 92.1 | 1.7 | 92.4 | 61.2 | 34.9 |
| | XGradCAM (0.8) | 93.3 | 62.9 | 33.0 | 92.4 | 18.4 | 80.4 | 92.8 | 50.5 | 46.6 | 92.7 | 50.9 | 46.1 | 93.3 | 81.5 | 13.3 | 92.9 | 32.1 | 65.8 |
| | GradCAM++ (0.6) | 74.7 | 91.2 | 0.9 | 74.4 | 89.6 | 3.3 | 76.2 | 92.9 | 0.8 | 74.8 | 91.3 | 1.7 | 75.0 | 92.7 | 0.8 | 75.0 | 87.1 | 6.6 |
| | GradCAM++ (0.7) | 85.1 | 90.8 | 2.4 | 84.7 | 83.6 | 10.4 | 85.7 | 93.1 | 0.8 | 84.9 | 92.0 | 1.5 | 85.5 | 93.0 | 0.9 | 85.3 | 75.7 | 19.0 |
| | GradCAM++ (0.8) | 90.8 | 66.0 | 29.5 | 90.0 | 51.4 | 45.1 | 90.7 | 77.5 | 17.5 | 90.3 | 82.3 | 12.2 | 90.8 | 92.4 | 1.5 | 90.5 | 49.9 | 46.7 |
| PatchCleanser | Vanilla | 89.9 | 37.2 | 60.7 | 89.7 | 46.2 | 53.0 | 90.0 | 47.3 | 52.2 | 89.9 | 49.9 | 48.7 | 90.1 | 48.6 | 50.8 | 90.0 | 34.3 | 65.0 |
| | Variant | 58.1 | 77.6 | 1.5 | 56.2 | 87.5 | 2.3 | 57.7 | 90.1 | 1.1 | 57.2 | 87.2 | 1.7 | 59.1 | 90.2 | 0.8 | 57.4 | 84.0 | 3.8 |
| Blur | Weak | 92.0 | 0.5 | 99.4 | 90.9 | 6.8 | 92.7 | 91.5 | 0.3 | 99.7 | 91.2 | 7.3 | 92.3 | 92.1 | 0.1 | 99.9 | 91.6 | 68.8 | 25.2 |
| | Strong | 65.6 | 65.0 | 3.5 | 61.5 | 59.9 | 5.6 | 63.8 | 62.6 | 3.4 | 63.3 | 54.0 | 14.8 | 64.9 | 60.1 | 6.7 | 62.4 | 58.6 | 4.4 |
| ShrinkPad | Weak | 91.5 | 44.7 | 51.3 | 89.6 | 53.3 | 42.6 | 89.6 | 6.3 | 93.2 | 90.8 | 82.2 | 9.6 | 91.5 | 28.3 | 70.1 | 91.1 | 86.9 | 3.1 |
| | Strong | 88.3 | 32.1 | 64.5 | 85.0 | 9.8 | 87.5 | 83.5 | 2.4 | 97.4 | 87.2 | 82.0 | 6.4 | 88.4 | 11.0 | 88.3 | 87.7 | 83.1 | 3.2 |
| Ours | Base | 93.0 | 90.7 | 0.6 | 91.8 | 90.3 | 1.5 | 92.5 | 91.5 | 0.5 | 92.3 | 90.0 | 1.5 | 92.9 | 92.1 | 0.5 | 92.6 | 90.5 | 0.8 |
| | Base-$i$ | 92.5 | 73.4 | 20.0 | 91.4 | 86.4 | 6.4 | 92.1 | 91.0 | 1.1 | 91.8 | 84.2 | 8.4 | 92.4 | 90.1 | 2.9 | 92.0 | 84.1 | 8.1 |
| | Base-$l$ | 92.1 | 91.0 | 0.5 | 90.6 | 90.4 | 1.4 | 91.6 | 91.4 | 0.9 | 91.1 | 90.1 | 1.5 | 92.1 | 91.0 | 1.7 | 91.5 | 90.2 | 0.8 |
| | Large | 93.1 | 90.8 | 0.6 | 92.0 | 90.6 | 1.4 | 92.7 | 91.8 | 0.5 | 92.5 | 90.4 | 1.3 | 93.1 | 92.3 | 0.5 | 92.7 | 90.9 | 0.8 |

Table A.4: Defense results on `GSTRB` using various sizes of color/white triggers.

| | | 1×1-color | | | 1×1-white | | | 2×2-color | | | 2×2-white | | | 3×3-color | | | 3×3-white | | |
|---|---|---|---|---|---|---|---|---|---|---|---|---|---|---|---|---|---|---|---|---|
| | | CA | BA | ASR | CA | BA | ASR | CA | BA | ASR | CA | BA | ASR | CA | BA | ASR | CA | BA | ASR |
| Before Defense | | 98.7 | 0.0 | 100. | 98.0 | 2.3 | 97.5 | 98.3 | 0.0 | 100. | 98.5 | 4.4 | 95.5 | 98.8 | 0.0 | 100. | 98.4 | 1.4 | 98.5 |
| Februus | XGradCAM (0.6) | 73.0 | 29.1 | 64.1 | 74.7 | 68.2 | 25.1 | 58.6 | 58.6 | 15.2 | 56.9 | 40.3 | 52.7 | 59.3 | 83.2 | 4.3 | 69.7 | 26.0 | 66.2 |
| | XGradCAM (0.7) | 85.2 | 22.9 | 74.6 | 84.8 | 59.1 | 38.1 | 74.8 | 47.4 | 41.9 | 74.4 | 17.6 | 81.2 | 74.8 | 77.0 | 16.3 | 82.4 | 7.9 | 90.8 |
| | XGradCAM (0.8) | 93.2 | 14.5 | 84.8 | 92.2 | 45.3 | 53.3 | 88.3 | 30.0 | 67.7 | 88.0 | 2.9 | 97.0 | 88.3 | 51.1 | 46.4 | 91.8 | 2.9 | 96.8 |
| | GradCAM++ (0.6) | 65.2 | 45.3 | 33.5 | 46.1 | 94.3 | 0.2 | 47.0 | 59.8 | 17.2 | 45.0 | 91.5 | 1.6 | 48.0 | 86.2 | 0.9 | 49.8 | 73.5 | 6.1 |
| | GradCAM++ (0.7) | 80.3 | 40.8 | 50.8 | 64.3 | 94.8 | 1.6 | 64.8 | 46.7 | 44.5 | 65.5 | 79.2 | 18.4 | 64.7 | 89.1 | 4.3 | 68.1 | 63.2 | 29.2 |
| | GradCAM++ (0.8) | 91.3 | 28.4 | 69.3 | 81.8 | 91.0 | 6.2 | 83.0 | 17.5 | 81.4 | 83.6 | 22.4 | 77.3 | 82.6 | 70.2 | 26.7 | 84.6 | 30.9 | 67.4 |
| PatchCleanser | Vanilla | 95.3 | 4.4 | 94.6 | 95.9 | 10.1 | 89.5 | 94.6 | 12.8 | 87.2 | 95.0 | 16.8 | 83.0 | 94.6 | 10.8 | 89.1 | 94.2 | 5.0 | 94.6 |
| | Variant | 12.7 | 45.9 | 0.5 | 12.7 | 88.8 | 0.8 | 14.4 | 97.8 | 0.1 | 13.8 | 93.3 | 2.3 | 14.9 | 96.1 | 0.0 | 11.3 | 63.1 | 5.5 |
| Blur | Weak | 98.6 | 3.0 | 96.9 | 98.0 | 9.4 | 90.3 | 98.2 | 0.1 | 99.9 | 98.5 | 6.6 | 93.3 | 98.6 | 0.2 | 99.8 | 98.3 | 4.1 | 95.6 |
| | Strong | 97.9 | 97.3 | 0.2 | 97.3 | 97.3 | 0.1 | 97.6 | 96.8 | 0.9 | 97.7 | 94.5 | 3.1 | 98.0 | 96.1 | 1.8 | 97.8 | 87.3 | 4.6 |
| ShrinkPad | Weak | 98.0 | 40.4 | 57.9 | 96.9 | 25.0 | 74.6 | 96.7 | 3.2 | 96.7 | 97.8 | 58.9 | 39.8 | 98.0 | 12.9 | 87.0 | 97.8 | 59.4 | 34.4 |
| | Strong | 94.1 | 27.7 | 69.4 | 91.0 | 5.1 | 94.2 | 91.8 | 2.2 | 97.8 | 92.9 | 32.8 | 64.5 | 93.7 | 10.4 | 89.0 | 93.3 | 62.9 | 18.6 |
| Ours | Base | 98.5 | 92.4 | 0.2 | 97.6 | 95.9 | 1.0 | 98.0 | 97.8 | 0.1 | 98.3 | 96.5 | 1.3 | 98.6 | 98.3 | 0.1 | 98.1 | 91.0 | 2.8 |
| | Base-$i$ | 73.0 | 29.1 | 64.1 | 74.7 | 68.2 | 25.1 | 58.6 | 58.6 | 15.2 | 56.9 | 40.3 | 52.7 | 59.3 | 83.2 | 4.3 | 69.7 | 26.0 | 66.2 |
| | Base-$l$ | 65.2 | 45.3 | 33.5 | 64.3 | 94.8 | 1.6 | 47.0 | 59.8 | 17.2 | 45.0 | 91.5 | 1.6 | 64.7 | 89.1 | 4.3 | 68.1 | 63.2 | 29.2 |
| | Large | 98.7 | 94.7 | 0.2 | 97.9 | 96.3 | 0.9 | 98.3 | 98.0 | 0.1 | 98.5 | 96.6 | 1.5 | 98.8 | 98.3 | 0.2 | 98.3 | 91.8 | 2.9 |

Table A.5: Defense results on `VGGFace2` using various types of triggers.

| | | Trigger '📷' | | | Trigger '💼' | | | Trigger '📌' | | | Trigger '🐦' | | | Trigger '🟢' | | |
|---|---|---|---|---|---|---|---|---|---|---|---|---|---|---|---|---|---|---|
| | | CA | BA | ASR | CA | BA | ASR | CA | BA | ASR | CA | BA | ASR | CA | BA | ASR |
| Before Defense | | 95.6 | 0.0 | 100. | 95.6 | 0.0 | 100. | 95.5 | 0.0 | 100. | 95.4 | 0.0 | 100. | 95.6 | 0.0 | 100. |
| Februus | XGradCAM | 65.2 | 94.1 | 0.0 | 65.3 | 95.0 | 0.0 | 67.5 | 95.4 | 0.0 | 64.6 | 68.2 | 28.6 | 65.0 | 94.8 | 0.0 |
| | GradCAM++ | 62.8 | 94.1 | 0.0 | 62.5 | 95.0 | 0.0 | 65.3 | 95.4 | 0.0 | 62.3 | 67.7 | 29.2 | 62.5 | 94.8 | 0.1 |
| PatchCleanser | Vanilla | 93.2 | 42.4 | 57.6 | 93.2 | 45.8 | 54.1 | 92.9 | 43.8 | 56.1 | 93.0 | 44.8 | 55.2 | 92.8 | 38.3 | 61.6 |
| | Variant | 51.1 | 94.3 | 0.0 | 51.7 | 94.6 | 0.0 | 49.6 | 95.3 | 0.0 | 50.6 | 95.1 | 0.0 | 50.3 | 94.3 | 0.0 |
| Blur | Weak | 95.5 | 0.0 | 100. | 95.5 | 0.1 | 99.9 | 95.5 | 0.1 | 99.9 | 95.3 | 0.1 | 99.9 | 95.5 | 0.0 | 100. |
| | Strong | 95.3 | 0.1 | 99.9 | 95.3 | 22.0 | 77.5 | 95.2 | 28.4 | 70.9 | 95.0 | 0.6 | 99.4 | 95.2 | 0.8 | 99.2 |
| ShrinkPad | Weak | 93.9 | 9.1 | 90.6 | 93.8 | 54.4 | 42.0 | 93.9 | 67.3 | 28.9 | 93.7 | 22.9 | 75.9 | 93.9 | 23.9 | 75.2 |
| | Strong | 88.4 | 35.2 | 61.1 | 88.2 | 71.5 | 17.7 | 88.5 | 78.6 | 11.1 | 88.1 | 42.8 | 50.7 | 88.2 | 43.7 | 51.1 |
| Ours | Base | 91.3 | 91.8 | 2.1 | 91.3 | 92.3 | 1.2 | 91.3 | 93.1 | 0.1 | 91.3 | 93.2 | 0.0 | 91.4 | 89.6 | 4.7 |
| | Large-$i$ | 91.1 | 68.0 | 28.1 | 91.1 | 79.0 | 16.2 | 91.0 | 89.4 | 4.8 | 91.0 | 71.4 | 23.7 | 91.2 | 52.9 | 44.1 |
| | Large-$l$ | 90.9 | 92.0 | 2.2 | 90.7 | 92.2 | 0.9 | 90.7 | 92.4 | 0.1 | 90.6 | 92.5 | 0.1 | 90.9 | 87.3 | 7.5 |
| | Large | 93.0 | 92.4 | 1.9 | 92.8 | 92.6 | 1.2 | 93.0 | 93.5 | 0.1 | 92.7 | 93.5 | 0.0 | 92.9 | 87.0 | 7.9 |

Table A.6: Defense results on `ImageNet10` using various types of triggers ([†]without adaptive thresholds adjustment).

| | | 1×1-color | | | 2×2-color | | | 3×3-color | | | Trigger '🖥' | | | Trigger '☂' | | | Trigger '🍉' | | |
|---|---|---|---|---|---|---|---|---|---|---|---|---|---|---|---|---|---|---|---|
| | | CA | BA | ASR | CA | BA | ASR | CA | BA | ASR | CA | BA | ASR | CA | BA | ASR | CA | BA | ASR |
| Before Defense | | 88.4 | 45.5 | 49.6 | 89.2 | 8.6 | 90.4 | 89.8 | 3.6 | 96.1 | 89.7 | 0.0 | 100. | 89.9 | 0.7 | 99.1 | 89.9 | 0.0 | 100. |
| PatchCleanser | Vanilla | 83.0 | 62.3 | 29.8 | 84.7 | 57.8 | 39.2 | 84.5 | 55.2 | 41.9 | 85.0 | 58.0 | 38.9 | 84.9 | 59.4 | 33.9 | 84.8 | 55.0 | 38.8 |
| | Variant | 55.4 | 67.2 | 14.2 | 63.8 | 84.4 | 3.6 | 62.3 | 84.6 | 2.8 | 62.2 | 87.5 | 1.4 | 64.7 | 79.3 | 1.9 | 63.9 | 82.0 | 0.9 |
| Blur | Weak | 87.3 | 49.6 | 44.1 | 88.7 | 18.8 | 79.4 | 88.5 | 16.1 | 82.0 | 88.1 | 0.1 | 99.9 | 88.8 | 1.7 | 98.1 | 89.1 | 0.2 | 99.8 |
| | Strong | 82.7 | 61.3 | 29.4 | 85.9 | 64.7 | 26.8 | 85.6 | 57.0 | 33.6 | 84.9 | 6.0 | 93.8 | 84.7 | 10.2 | 88.6 | 84.9 | 6.2 | 93.3 |
| ShrinkPad | Weak | 87.7 | 80.7 | 9.0 | 87.8 | 76.9 | 14.3 | 88.7 | 60.2 | 33.1 | 89.0 | 6.9 | 92.7 | 88.5 | 28.9 | 68.0 | 88.8 | 4.3 | 95.6 |
| | Strong | 85.8 | 82.6 | 6.0 | 86.4 | 86.1 | 2.5 | 86.9 | 78.2 | 11.2 | 87.1 | 40.8 | 56.2 | 86.5 | 39.5 | 54.2 | 87.5 | 13.1 | 85.7 |
| Ours[†] | Base | 65.1 | 60.4 | 17.5 | 71.6 | 70.8 | 7.0 | 74.4 | 73.1 | 4.6 | 73.3 | 75.9 | 3.2 | 73.3 | 69.0 | 4.5 | 72.8 | 75.4 | 2.2 |
| | Large-$i$ | 65.2 | 56.8 | 25.6 | 70.3 | 54.8 | 33.8 | 71.6 | 58.5 | 28.2 | 72.9 | 68.3 | 13.7 | 72.7 | 68.4 | 12.8 | 73.4 | 66.7 | 15.5 |
| | Large-$l$ | 83.8 | 75.8 | 11.8 | 85.6 | 81.2 | 5.8 | 86.2 | 85.1 | 2.7 | 86.8 | 87.3 | 1.2 | 85.9 | 84.3 | 1.7 | 86.4 | 87.8 | 0.7 |
| | Large | 77.2 | 69.9 | 13.5 | 81.8 | 77.7 | 5.4 | 82.6 | 80.0 | 3.4 | 82.5 | 80.5 | 1.6 | 82.5 | 73.3 | 3.8 | 83.0 | 80.1 | 1.4 |
| Ours | Base | 75.1 | 71.4 | 13.8 | 80.1 | 80.8 | 5.4 | 80.9 | 83.8 | 3.1 | 81.5 | 84.1 | 2.5 | 81.0 | 82.7 | 2.4 | 80.9 | 83.7 | 1.7 |
| | Large-$i$ | 79.0 | 62.0 | 27.4 | 82.8 | 49.9 | 43.9 | 82.8 | 53.4 | 39.7 | 83.8 | 61.2 | 28.7 | 83.7 | 67.2 | 20.5 | 83.3 | 59.0 | 31.0 |
| | Large-$l$ | 84.0 | 76.7 | 11.8 | 85.5 | 82.2 | 5.9 | 86.3 | 85.7 | 2.6 | 86.8 | 87.3 | 1.2 | 86.0 | 85.0 | 1.6 | 86.6 | 87.7 | 0.7 |
| | Large | 81.4 | 76.2 | 10.9 | 84.0 | 83.2 | 5.6 | 84.2 | 84.9 | 2.9 | 84.6 | 86.9 | 1.5 | 84.1 | 84.7 | 1.4 | 85.1 | 86.4 | 1.0 |

Table A.7: Defense results on `ImageNet50` using various types of triggers.

| | | 1×1-color | | | 2×2-color | | | 3×3-color | | | Trigger '🖥' | | | Trigger '☂' | | | Trigger '🍉' | | |
|---|---|---|---|---|---|---|---|---|---|---|---|---|---|---|---|---|---|---|---|
| | | CA | BA | ASR | CA | BA | ASR | CA | BA | ASR | CA | BA | ASR | CA | BA | ASR | CA | BA | ASR |
| Before Defense | | 83.7 | 2.4 | 97.0 | 83.8 | 0.2 | 99.7 | 83.9 | 0.2 | 99.8 | 84.4 | 0.0 | 100. | 84.1 | 0.1 | 99.9 | 84.1 | 0.0 | 100. |
| PatchCleanser | Vanilla | 79.1 | 47.1 | 48.2 | 79.4 | 44.9 | 51.7 | 79.8 | 44.4 | 52.8 | 79.6 | 49.0 | 47.0 | 80.2 | 44.7 | 47.7 | 79.6 | 44.1 | 49.2 |
| | Variant | 52.9 | 79.6 | 1.1 | 53.8 | 81.3 | 0.3 | 54.3 | 81.8 | 0.4 | 54.8 | 82.1 | 0.2 | 54.2 | 75.1 | 0.1 | 54.6 | 76.2 | 0.1 |
| Blur | Weak | 82.9 | 11.0 | 86.9 | 83.3 | 7.9 | 90.4 | 83.0 | 9.0 | 89.8 | 83.5 | 0.1 | 99.9 | 83.4 | 0.9 | 99.0 | 83.5 | 0.4 | 99.5 |
| | Strong | 78.5 | 62.5 | 21.1 | 79.3 | 75.5 | 5.3 | 78.9 | 64.6 | 19.2 | 79.6 | 11.2 | 87.2 | 79.5 | 38.3 | 54.4 | 79.7 | 42.6 | 48.6 |
| ShrinkPad | Weak | 81.7 | 70.2 | 13.9 | 81.9 | 77.1 | 6.1 | 81.9 | 49.8 | 38.5 | 82.3 | 7.5 | 91.4 | 82.2 | 27.8 | 63.5 | 82.2 | 6.1 | 93.2 |
| | Strong | 79.0 | 76.7 | 3.1 | 79.3 | 76.0 | 4.0 | 79.5 | 72.4 | 7.9 | 79.9 | 41.9 | 48.5 | 79.7 | 46.1 | 36.5 | 79.3 | 17.7 | 78.5 |
| Ours[†] | Base | 52.8 | 54.9 | 1.6 | 53.6 | 61.1 | 0.8 | 52.8 | 60.4 | 0.9 | 52.7 | 59.9 | 1.3 | 51.7 | 48.4 | 1.5 | 52.5 | 56.7 | 1.0 |
| | Large-$i$ | 57.2 | 48.5 | 22.7 | 57.2 | 37.1 | 44.8 | 57.6 | 47.4 | 29.5 | 57.4 | 57.1 | 10.8 | 56.9 | 53.9 | 10.8 | 57.7 | 50.2 | 17.2 |
| | Large-$l$ | 75.0 | 75.5 | 1.4 | 75.7 | 77.3 | 0.4 | 75.9 | 77.8 | 0.3 | 75.7 | 81.0 | 0.2 | 76.2 | 74.3 | 0.2 | 75.8 | 78.6 | 0.2 |
| | Large | 66.5 | 62.7 | 1.7 | 67.2 | 69.7 | 0.7 | 67.5 | 69.1 | 0.6 | 66.9 | 67.7 | 0.6 | 67.0 | 56.5 | 0.9 | 67.0 | 64.3 | 0.6 |
| Ours | Base | 61.8 | 69.9 | 1.4 | 62.3 | 74.6 | 0.5 | 61.8 | 74.0 | 0.6 | 61.7 | 69.8 | 0.8 | 61.3 | 65.5 | 0.6 | 61.4 | 66.7 | 0.8 |
| | Large-$i$ | 71.7 | 48.3 | 34.1 | 71.7 | 35.8 | 52.9 | 72.1 | 47.6 | 38.1 | 71.4 | 55.9 | 24.8 | 72.0 | 56.7 | 19.7 | 71.9 | 48.1 | 32.7 |
| | Large-$l$ | 75.7 | 77.0 | 1.4 | 76.5 | 79.1 | 0.3 | 76.4 | 79.4 | 0.3 | 76.3 | 81.0 | 0.2 | 76.8 | 75.9 | 0.2 | 76.4 | 78.7 | 0.2 |
| | Large | 72.2 | 75.3 | 1.3 | 72.7 | 78.3 | 0.5 | 73.0 | 78.7 | 0.4 | 72.7 | 76.9 | 0.4 | 72.4 | 73.2 | 0.4 | 72.6 | 74.2 | 0.5 |

Table A.8: Defense results on `ImageNet100` using various types of triggers.

| | | 1×1-color | | | 2×2-color | | | 3×3-color | | | Trigger '🖥' | | | Trigger '☂' | | | Trigger '🍉' | | |
|---|---|---|---|---|---|---|---|---|---|---|---|---|---|---|---|---|---|---|---|
| | | CA | BA | ASR | CA | BA | ASR | CA | BA | ASR | CA | BA | ASR | CA | BA | ASR | CA | BA | ASR |
| Before Defense | | 81.9 | 0.9 | 98.9 | 82.4 | 0.1 | 99.9 | 82.6 | 0.1 | 99.9 | 82.4 | 0.0 | 100. | 82.3 | 0.0 | 100. | 82.4 | 0.0 | 100. |
| PatchCleanser | Vanilla | 78.5 | 44.1 | 51.9 | 78.9 | 43.2 | 53.6 | 79.2 | 43.3 | 53.7 | 79.0 | 46.5 | 49.9 | 78.8 | 41.1 | 52.5 | 78.9 | 42.3 | 52.1 |
| | Variant | 51.0 | 78.9 | 0.3 | 51.9 | 80.1 | 0.1 | 52.8 | 80.7 | 0.1 | 52.5 | 80.5 | 0.1 | 52.0 | 72.7 | 0.1 | 52.2 | 75.3 | 0.1 |
| Blur | Weak | 80.7 | 6.4 | 92.2 | 81.1 | 6.1 | 92.7 | 81.3 | 5.9 | 93.0 | 81.2 | 0.1 | 99.9 | 81.2 | 0.8 | 99.2 | 81.4 | 0.2 | 99.7 |
| | Strong | 75.5 | 67.5 | 10.7 | 75.9 | 74.1 | 2.4 | 76.2 | 62.6 | 18.3 | 75.9 | 24.2 | 71.0 | 76.4 | 37.1 | 53.3 | 76.2 | 44.3 | 43.1 |
| ShrinkPad | Weak | 79.7 | 66.4 | 16.2 | 79.9 | 74.9 | 6.4 | 80.2 | 54.8 | 29.6 | 80.1 | 11.6 | 86.6 | 79.9 | 37.4 | 47.7 | 79.9 | 8.9 | 89.7 |
| | Strong | 76.7 | 74.6 | 2.3 | 77.0 | 74.2 | 3.5 | 77.3 | 71.5 | 7.1 | 77.4 | 53.5 | 31.2 | 77.7 | 52.4 | 22.8 | 77.0 | 25.6 | 68.0 |
| Ours[†] | Base | 48.4 | 52.3 | 0.8 | 49.8 | 58.6 | 0.5 | 49.4 | 58.4 | 0.6 | 49.4 | 57.6 | 0.8 | 48.9 | 46.3 | 0.8 | 48.5 | 53.6 | 0.6 |
| | Large-$i$ | 54.4 | 47.0 | 22.0 | 54.7 | 37.1 | 43.1 | 54.7 | 46.0 | 28.8 | 54.4 | 54.1 | 11.2 | 54.2 | 51.1 | 10.8 | 54.2 | 47.9 | 16.4 |
| | Large-$l$ | 71.8 | 73.8 | 0.6 | 72.1 | 75.4 | 0.1 | 72.1 | 76.0 | 0.1 | 72.1 | 79.3 | 0.1 | 72.0 | 71.4 | 0.1 | 71.9 | 76.3 | 0.1 |
| | Large | 62.9 | 60.0 | 0.8 | 63.3 | 67.3 | 0.3 | 63.2 | 65.3 | 0.4 | 62.8 | 64.3 | 0.4 | 62.7 | 53.6 | 0.7 | 62.9 | 61.0 | 0.5 |
| Ours | Base | 58.5 | 67.3 | 0.6 | 59.6 | 73.1 | 0.2 | 59.1 | 71.9 | 0.3 | 59.3 | 68.1 | 0.5 | 58.7 | 62.9 | 0.3 | 58.7 | 64.0 | 0.6 |
| | Large-$i$ | 68.8 | 47.2 | 33.7 | 69.3 | 36.3 | 51.2 | 68.9 | 46.3 | 37.8 | 69.1 | 53.7 | 25.1 | 69.1 | 54.7 | 19.9 | 68.9 | 46.4 | 31.8 |
| | Large-$l$ | 73.2 | 75.4 | 0.6 | 73.4 | 77.4 | 0.1 | 73.4 | 77.6 | 0.1 | 73.6 | 79.3 | 0.1 | 73.3 | 73.1 | 0.1 | 73.2 | 76.3 | 0.1 |
| | Large | 69.4 | 73.8 | 0.5 | 69.6 | 76.9 | 0.2 | 69.5 | 76.6 | 0.2 | 69.4 | 73.9 | 0.3 | 69.3 | 70.8 | 0.2 | 69.4 | 71.4 | 0.4 |

