# OpenReview forum: "Mask and Restore: Blind Backdoor Defense at Test Time with Masked Autoencoder"
_ICLR.cc/2024/Conference — ICLR 2024 Conference Withdrawn Submission_

### Official Review · Reviewer_mNMh · 2023-10-29

**Soundness:** 3 good
**Presentation:** 2 fair
**Contribution:** 2 fair
**Rating:** 5
**Confidence:** 4

**Summary:**

This paper explores how to purify poisoned testing samples (with local triggers) based on a pre-trained masked auto-encoder (MAE) under the black-box setting where defenders can only query the deployed model and obtain its predictions. Specifically, the authors randomly generate MAE masks and calculate two trigger scores based on image structural similarity and label prediction consistency between test images and MAE restorations, respectively. The authors also exploit trigger topology to refine them further. After that, the authors will use their average to refine the poisoned testing sample. The authors evaluate their methods on CIFAR, VGGFace, and ImageNet dataset under 4 baseline defenses.

**Strengths:**

1. Black-box image purification is a practical setting for backdoor defenses.
2. The proposed method is easy to follow.
3. The evaluation is extensive under their primary focus to a large extent.

**Weaknesses:**

1. One of my biggest concerns is its setting. To the best of my knowledge, most of the existing advanced backdoor attacks (e.g., WaNet and ISSBA) are with full-image-size triggers instead of local ones. However, the proposed defense can only remove local triggers. As such, its application is limited. Of course, I note that the authors explicitly state this in their limitations chapter, which is to be encouraged.
2. For me, the technical contributions are minor to some extent. Given we only consider local triggers, the main pipeline (without topology refinement) is straightforward. This part didn't enlighten me in any new way.
3. As for the topology refinement (Section 4.3), the authors should explicitly mention that there are many existing backdoor defenses (e.g., [1-2]) having a similar idea. The authors should also state the similarities and differences between their method and existing ones in the appendix.
4. I think there are many hyper-parameters involved in this paper, although the authors did not explicitly mention it. For example, those values in first two paragraphs (page 6) and in the second paragraph (Section 4.4). The authors should also discuss their effects.
5. Please provide more justifications of the last paragraph in Section 4.2.
6. Please provide more justifications of the last two sentences in Discussion and Limitation (page 9).
7. The ASR after defense is still high (e.g., >10%) in many cases.


References
1. Trigger Hunting with a Topological Prior for Trojan Detection.
2. Topological Detection of Trojaned Neural Networks.

**Questions:**

I would like to know whether the proposed method is still effective under the all-to-all setting.

---

> ### Author Response · Authors · 2023-11-21
> **Response to Reviewer mNMh (part I)**
>
> > Setting with local triggers.
>
> Our work focuses on local triggers. Local triggers are easy to be implemented in the physical world, for example by adding a sticker to the image. Meanwhile, they are robust to photometric noises or geometric distortions. These unique properties indicate that backdoor defense methods designed for (invisible) full-image-size triggers may not be optimal for local triggers. In the paper, we show that test-time image transformations (including the advanced ones with diffusion models) do not work well. Research on test-time defense for local triggers is still meaningful.
>
> We use the word 'local' as opposed to full-image-size. However, we do not restrict our work to small squared patches. In the paper, we consider several complex triggers. IAB uses sample-specific irregular cuerves. LC uses checkerboards at four distant corners. Blended uses an invisible patch. We design our method to be generalizable enough to handle all these cases.
>
>
> > Technical contributions.
>
> As explained in the previous question, we try to handle diverse triggers within the realm of local ones. Given that we are solving test-time defense for black-box models where only one single test image and its hard label predictions are available, a unified solution for this task is not straightforward. In the tables, we compare with PatchCleanser that uses a two step masking. Its assumption that the entire trigger can be covered by a small rectangle mask is not applicable to LC and IAB attacks. Test-time image transformations also could not work well.
>
> Our method simultaneously locate triggers and recover true labels. We innovatively integrate MAE into our method, and transform trigger detection into refining trigger scores in the token space. With the help of MAE reconstruction, we can successfully purify backdoored images with complex triggers like irregular curves without harming accuracies on clean images.
>
> > Topology refinement.
>
> We would like to thank for pointing out these two interesting works. 'Topology' is an important and basic concept that encodes prior knowledge on backdoor triggers and networks. (R1 Trigger Hunting) is related to ours but for a different target. It considers the topology of triggers during the reverse engineering to identify Trojaned models. The authors explicitly apply a topological loss during the optimization of trigger masks. Differently, our work uses the idea of topology to sample better MAE masks. We iteratively generate topology-aware MAE masks based on trigger scores, and use MAE restorations to refine trigger scores. (R2 Topological Detection) is not relevant to ours as it considers the topology of neural networks rather than triggers.
>
> > Hyper-parameters.
>
> Our method consists of several hyper-parameters. But we do not tune them for different tasks or datasets. Instead, we delicately designed our method so that the default hyper-parameters can be applied universally.
>
> In Sec. 4.4, we use a set of $\tau_k$ as the initial thresholds and adjust them automatically during defense. As shown in Fig.5, the range of optimal thresholds is large after topology-aware refinement, thus the method is not sensitive to the choice of thresholds. In Sec. F.1-F.2, we include analysis on the topology sampling hyper-parameter and repeated times. The default values achieve a balance between performances and computation cost.

---

> > ### Comment · Reviewer_mNMh · 2023-11-23
> >
> > 1. It is okay only to consider defending against attacks in the physical world. However, you should explicitly mention it in your paper (or probably even in your title) instead of trying to hide it. IAB and four-corner triggers are still local even though they are not trigger patches. Let me try to explain it more clearly: we can still use a few patches to cover their triggers. This is why I want you to evaluate under the WaNet or ISSBA.
> > 2. I value your studied problem. However, as I mentioned, the main pipeline (without topology refinement) is straightforward if we only consider local triggers. Honestly, I cannot learn new things from it. This is why I still believe it is with minor technical novelty. However, let me re-emphasize: I think you've researched an interesting problem (but you should carefully revise your statement to avoid overclaim).
> > 3. You should cite them in your main paper and compare them in detail in the appendix.
> > 4. Yes, you did provide the ablation study about hyper-parameters defined in this paper. But I intended to mention that there are still some (or even many) hyper-parameters that you simply assigned a value (e.g., 0.5, 25%) without ablation study.
> > 5. My question is well addressed.
> > 6. My question is well addressed.
> > 7. I fully understand the difficulty of this task. I will reserve my opinion on this one, but it won't affect my scores.
> > 8. Thank you and good to know it.
> >
> > I slightly increase my score since the author have addressed some of my concerns.

---

> > > ### Author Response · Authors · 2023-11-23
> > >
> > > We would like to thank the reviewer and appreciate the followup comments. Regarding 1, we respectfully make some clarifications on the possible misunderstandings from previous response.
> > >
> > > Our work is not meant for defending in the physical world. We mentioned "Local triggers are easy to be implemented in the physical world" just as an example to show that local triggers are practical and useful. Our work focuses on general local triggers. We explicitly mentioned this in the main text.
> > >
> > > Regarding IAB and LC, it's true that "we can still use a few patches to cover their triggers" (and certainly this is what local triggers should be like). But how to efficiently find those covering patches is still unknown. In particular, we need to cover triggers only but not clean regions (otherwise the predicted label will be incorrect). This is a combination problem if we don't have any prior knowledge on the shape and location of triggers.
> > >
> > > One main contribution, by leveraging MAE, we reduce the search to token space and refine the trigger mask (scores) with structural similarity, label consistency and topology prior. Our method takes about 10 times refinements for the trigger mask (scores) to converge.
> > >
> > > We appreciate the reviewer's suggest on WaNet or ISSBA. However, as explained in the previous response, our method is not designed for global triggers. For such triggers, it is hard to define a mask that can cover triggers without affecting clean regions. Instead, we think image transformation based defenses such as adding noises or distortion should be more appropriate. Due to different properties of local and global triggers, we did not intend to adapt our methods to global triggers but prefer to build defenders separately.

---

> ### Author Response · Authors · 2023-11-21
> **Response to Reviewer mNMh (part II)**
>
> > Justifications of the last paragraph in Section 4.2.
>
> The image-based score leverages the structural similarity between original image and its MAE restorations. When triggers are large and complex, the SSIM scores in the trigger regions are more salient, thus it is easy to detect such trigger. Otherwise when triggers are small, the structural difference between original image and MAE restorations are small, making it hard to detect triggers.
>
> The label-based score works in an opposite manner. When triggers are small, it is likely that MAE masks can remove the triggers, thus the label prediction changes to the ground-truth. Otherwise when triggers are large, some parts of the triggers may remain in the MAE restorations, thus the model still predict test image as the target label.
>
> In Sec. F.4, we have included related discussions on varying trigger size and the corresponding different behaviours of two scores.
>
> > Justifications of the last two sentences in Discussion and Limitation.
>
> Our work focuses on local triggers. For invisible non-local triggers, we can apply an additional test-time image transformation to the test image. Although these transformations could not work on visible local triggers, they are likely to work on invisible global triggers. In (Shi et al., 2023), the authors deal with a similar task as ours. They first add some random noises to the image, and then use diffusion models to restore image. It's hard to incapacitate complex triggers (like IAB curves) using some global transformation without harming clean accuracies. As discussed in Sec.3, detecting the location of possible local triggers is still important. We believe the combination of test-time image transformation and our method is an interesting work to handle both local and non-local triggers.
>
> > ASR after defense.
>
> We highlight that the studied task of test-time defense for black-box models is quite challenging. Only the single test image and its hard label predictions are available. Some high ASR occurs on challeging attacks like IAB. The corresponding triggers are frequently long curves going through the whole image or a couple of fragments. In these cases, the comparison methods completely fail, while ours obtain a decent result. Also, we consider both accuracies on clean image and backdoored images. Our method is not tuned to get the best ASR scores at the expense of clean accuracies.
>
> > Effective under the all-to-all setting.
>
> We conducted IAB attack with all-to-all setting. The comparison results are list below. Our method still works the best in such setting.
> | Method |  CA | BA | ASR |
> | ---- | ---- | ---- | ---- |
> |Before Defense | 93.59	| 2.10	| 91.11 |
> |PatchCleanser (Vanilla) | 88.80	| 18.64	| 73.00 |
> |PatchCleanser (Variant)  | 61.47	| 49.83	| 23.42|
> |Blur (Weak) | 91.45	| 35.61	| 55.22 |
> |Blur (Strong) | 63.73	| 56.02	| 3.88 |
> |ShrinkPad (Weak) | 91.25	| 74.45	|15.38|
> |ShrinkPad (Strong) | 88.33	| 84.14	| 2.63 |
> |Ours (Base) | 93.12	| 85.08	| 4.84 |
> |Ours (Large) | 93.33	| 84.81	| 5.90 |

---

### Official Review · Reviewer_j6HB · 2023-11-01

**Soundness:** 3 good
**Presentation:** 2 fair
**Contribution:** 2 fair
**Rating:** 3
**Confidence:** 4

**Summary:**

This paper proposes to leverage Masked AutoEncoder (MAE) to remove backdoor on samples at test time. For detecting the triggered patches, it proposes the image-based score and the label-based score to guide the masking.

**Strengths:**

1. This paper proposes a test-time backdoor defense method. Test-time backdoor defense is an important and vital topic.

**Weaknesses:**

1. The motivation is unclear. The introduction of MAE will cause the model to take much time during inference. There are lots of well-performed backdoor sample detection methods that can detect backdoor samples accurately. From my point of view, it is good enough to detect the poisoned samples. Why is it necessary to give the ground-truth label of the triggered samples?
2. I wonder why DDPM-based method has only around 70% accuracy in Table 1. I think it is strange and can the authors provide a reason?
3. The number of model architectures and attack methods being tested is clearly below the ICLR acceptance threshold. All the experiments are conducted on ResNet against only 4 attacks.
4. Model repairing-based backdoor defense should be compared.  ACC and ASR in Table 3 after the backdoor defense BDMAE is not as good as the model repairing-based backdoor defense.

**Questions:**

Does MAE need to be finetuned when employed to CIFAR10 or CIFAR100?

---

> ### Author Response · Authors · 2023-11-21
> **Response to Reviewer j6HB**
>
> > Motivation of introducing MAE. Other backdoor sample detection methods.
>
>  We highlight that the task of blind backdoor defense at test time for black-boxed models is practical yet challenging. Our goal is to recover the true labels, rather than detecting backdoor samples but refuse to make a decision. For a real system, a practical defense module is expected to return true labels regardless of inputs. Otherwise, the system service would be interrupted.
>
> To the best of our knowledge, existing methods either requires knowing model details or additional validation data, or could not make correct predictions for both clean and backdoored images.
>
> We show in the paper that test-time image transformation could not perform well. Naive trigger search in the image space could not generalize to complex triggers. Therefore, we are motivated to locate possible triggers and recover true labels simultaneously. The integration of MAE enable us to handle a variety of triggers in a unified manner. MAE is only used for inference without any updates. The increased overhead in inference time is still much lower than using diffusion models.
>
> > Performance of DDPM-based method.
>
> We have made discussions in Sec. 3 and Sec. F.7 about the fundamental differences between ours and diffusion model-based method (DiffPure). DiffPure first adds a small amount of noise to the image and then uses a reverse generative process to recover the clean images. There are a few reasons about its unsatisfactory performances for backdoor image purification.
> 1) It applies a global transformation to the entire image, without specifically locating and removing triggers. The triggers may partially remain in the restored images, as shown in Fig. A.6-A.7. Thus the resulted images may still be predicted as the target label, leading to a high ASR and low BA.
> 2) Diffusion models are pretrained on a large resolution. For Cifar10 and GTSRB of size 32$\times$32, the diffusion model restored images after down-scaling may not be consistent with the original image in the clean regions.
> 3) Diffusion models could hallucinate some details not existing in the original images. For example, in the VGGFace2 dataset, features like eye-glasses, expressions, mouths are changed with diffusion models. These features are critical to recognizing face identity.
>
> The last two reasons explain why clean accuracies are low for diffusion model-based defense.
>
>
> > Number of model architectures and attack methods being tested.
>
>  Please note that we are dealing with black-box models. Therefore, the model architectures should not affect the conclusions. That being said, we have provided results with a simple convolution network used in previous work and VGG16 backbone in Sec. F.3.
>
> Since we work on test-time image purification, the diversity of triggers is more important than the number of attack methods. In the experiments, we evaluate our method against a variety of triggers, including different sizes of white patches and color patches, commonly seen icons, irregular curves, four distant checkerboards and invisible patches. We believe they are representative enough for local triggers.
>
> > Model repairing-based backdoor defense.
>
>  We highlight that our task is test-time defense for black-box models. Only one single test image and its hard label predictions are available. Model repairing-based backdoor defense is not applicable to our task. Their results cannot be directly compared with ours due to different settings.
>
> > MAE finetuning.
>
> No, MAE is not updated for all experiments. Since only one test image (and possibly backdoored) is available, we believe there is no particular benefits in fine-tuing MAE.

---

### Official Review · Reviewer_9rDY · 2023-11-10

**Soundness:** 2 fair
**Presentation:** 3 good
**Contribution:** 2 fair
**Rating:** 3
**Confidence:** 3

**Summary:**

The author proposes to use masked auto encoders (MAEs) to locate possible trigger patterns, and apply removal and reconstruction with MAEs to mask and restore the image in order to remove triggers. Using the restored images to train models can defend them effectively against backdoors.

**Strengths:**

* The method is in general agnostic to the triggers and models, which makes it more applicable than other approaches.
* The results show that the use of MAEs to defend against backdoors is effective.

**Weaknesses:**

* The time complexity involved with using a MAE to cleanse data is not well explained. Is the additional compute worth the effort?
* Many attacks generate stealthy but non-localized triggers, for instance, WaNet [1], LIRA [2], Marksman attacks [3], Flareon [4], etc. It remains to be seen how the proposed defense can be effective against such attacks.
* This paper assumes the availability of an MAE for image restoration, such models may not exist for the training dataset. In addition, despite mentioning the use of out-of-distribution (OOD) data to cleanse backdoors, no results are presented for OOD scenarios.

[1]: WaNet -- Imperceptible Warping-based Backdoor Attack, ICLR 2021, https://arxiv.org/abs/2102.10369

[2]: LIRA: Learnable, Imperceptible and Robust Backdoor Attacks, CVPR 2021, http://openaccess.thecvf.com/content/ICCV2021/papers/Doan_LIRA_Learnable_Imperceptible_and_Robust_Backdoor_Attacks_ICCV_2021_paper.pdf

[3]: Marksman backdoor: Backdoor attacks with arbitrary target class, NeurIPS 2022, https://proceedings.neurips.cc/paper_files/paper/2022/file/fa0126bb7ebad258bf4ffdbbac2dd787-Paper-Conference.pdf

[4]: Flareon: Stealthy any2any Backdoor Injection via Poisoned Augmentation, https://arxiv.org/abs/2212.09979

**Questions:**

* How likely is the model learning from the MAE instead of the training data? Is it possible to design experiments to find out?

---

> ### Author Response · Authors · 2023-11-21
> **Response to Reviewer 9rDY**
>
> > Time complexity and benefits using MAE.
>
>  Please refer to the general response.
>
>
> > About stealthy but non-localized triggers.
>
> Our work is tailored for the practical and important local triggers. Stealthy but non-localized triggers have different properties. The two types of triggers need to be handle separately. Stealthy triggers are more sensitive to image transformations. Differently for local triggers, how to detect their location is critical. In our proposed BDMAE, we leverage MAE to obtain trigger scores. They are not intended for non-localized triggers that span over the entire image.
>
> > Availability of MAE and out-of-distribution data.
>
> We use the generic pretrained MAE to assist backdoor purification, which is strong enough to transfer to different downstream datasets. Our task is test-time defense. Out-of-distribution data are not related to the task.
>
> > Model learning from the MAE instead of the training data.
>
> Our task is test-time defense. Each test image is purified on-the-fly. We don't use training data nor update MAE.

---

### Author Response · Authors · 2023-11-21
**General Response**

We thank reviewers for their time and feedback. We make general clarifications on our task and contributions here, and include detailed responses below.

- Our paper focuses on local triggers. The enthusiasm on stealthy but non-localized triggers does not nullify research on local triggers. In fact, we believe both types are important but with different properties that need to be handled separately. Local triggers have their merits like easy implementation in the physical world and robustness to noises or distortions. They cannot be defended with general test-time image transformations as we show in the paper. This is why we aim to locate triggers within images and recover true labels simultaneously.

- We handle the task of blind test-time defense for black-box models. It is practical considering that many models are black-box due to increasing concerns on privacy. Admittedly, this is quite challenging as only the single test image and its hard-label predictions can be used. This important setting has not been well explored. Existing methods that require model details and additional validation data are not applicable.

- We innovatively incorporate MAE to solve this challenging task. In the paper, we consider diverse triggers such as local icons of BadNets, sample-specific irregular curves of IAB, distant checkerboards of LC, invisible patches of Blended. Our method handles these triggers in a unified way, without assumptions on shapes, positions, etc. The use of MAE, while introducing some overhead, is still much more efficient than diffusion model-based defense.